# CodeQuant: Unified Clustering and Quantization for Enhanced Outlier Smoothing in Low-Precision Mixture-of-Experts

**Xiangyang Yin**[1*] **Xingyu Liu**[2*] **Tianhua Xia**[2] **Bo Bao**[3] **Vithursan Thangarasa**[3]
**Valavan Manohararajah**[3] **Eric Sather**[3] **Sai Qian Zhang**[1,2]
[1]Courant Institute of Mathematical Sciences, New York University
[2]Tandon School of Engineering, New York University
[3]Cerebras Systems Inc.
`{shawn.yin, xl5444, tx856, sai.zhang}@nyu.edu`
`{bo.bao, vithu, valavan, eric.sather}@cerebras.net`

## Abstract

Outliers have emerged as a fundamental bottleneck in preserving accuracy for low-precision large models, particularly within Mixture-of-Experts (MoE) architectures that are increasingly central to large-scale language modeling. Under post-training quantization (PTQ), these outliers induce substantial quantization errors, leading to severe accuracy degradation. While recent rotation-based smoothing techniques alleviate the problem by redistributing outlier magnitudes, residual errors remain and continue to impede reliable low-precision deployment.

In this work, we tackle this challenge by introducing *CodeQuant*, a unified quantization-and-clustering scheme that contains smoothing activation outliers via learnable rotation and absorbing weight outliers into fine-tuned cluster centroids for MoE. This design reduces the influence of extreme values by fitting them within cluster centroids, thereby lowering quantization error while maintaining expressive capacity. Coupled with a dedicated kernel design for GPU and CPU, CodeQuant achieves up to $4.15\times$ speedup while delivering significantly higher accuracy than state-of-the-art quantization approaches across diverse MoE models. Our results highlight CodeQuant as a promising direction for efficient and accurate deployment of MoE-based large language models under low-precision constraints. Our code is available at `https://github.com/SAI-Lab-NYU/CodeQuant`.

## 1 Introduction

Mixture-of-Experts (MoE) has emerged as one of the most effective paradigms for scaling large language models (LLMs). By activating only a subset of experts for each input token, MoE introduces conditional computation, allowing different experts to specialize in distinct linguistic or multimodal patterns. This specialization enables MoE-based models to achieve superior performance across diverse tasks. Consequently, MoE architectures have been adopted in many state-of-the-art LLMs (Abdin et al., 2024; Yang et al., 2025; DeepSeek-AI et al., 2024). Despite these advantages, MoE models still carry substantial computational and system-level costs. Although only a fraction of experts is active per token, the total parameter size is extremely large, leading to high memory requirements and increased communication overhead during distributed training and inference. These factors increase processing latency and pose serious challenges for real-world deployment.

To address these costs, low-precision quantization has become a widely adopted strategy. By representing weights and activations with fewer bits, quantization substantially reduces memory footprint and improves computational throughput. Recent hardware innovations further accelerate this trend: NVIDIA's Hopper (NVIDIA Corporation, 2022b) and Ada GPUs (NVIDIA Corporation, 2022a) natively support FP8 arithmetic, while the Blackwell series extends support to FP4. These developments provide a strong foundation for efficient MoE deployment with low precision. However,

---

*Equal contributions.

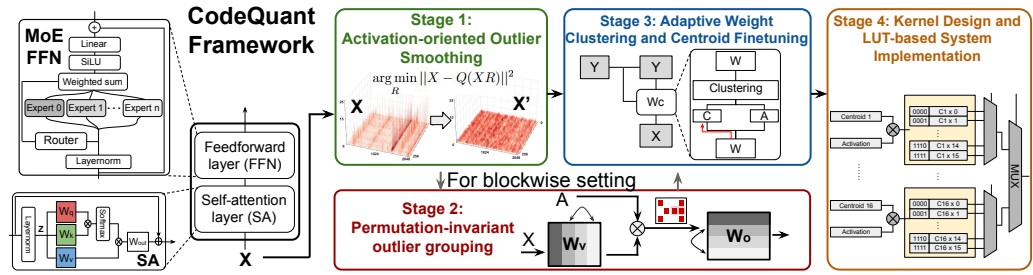

Figure 1: Overview of the CodeQuant framework. The left panel illustrates the target architectures, including MoE FFN and Self-Attention blocks. The right panel depicts the four-stage calibration and deployment pipeline: Stage 1 applies learnable rotations to smooth activation outliers; Stage 2 permute weight for optimized distribution; Stage 3 introduces clustering fine-tune mechanism to align with objective; and Stage 4 deploys the quantized model using a specialized LUT kernel.

quantizing MoE architectures remains challenging due to the prevalence of outliers (Dettmers et al., 2022; Sun et al., 2024). Large-magnitude activations expand the dynamic range, leading to severe quantization errors and significant accuracy degradation under post-training quantization (PTQ), particularly in low-bit settings such as 4-bit quantization. While recent outlier-smoothing methods (Xiao et al., 2024; Ashkboos et al., 2024) alleviate the issue, residual errors persist and continue to hinder reliable low-precision deployment.

In parallel, codebook-based approaches such as clustering have emerged as a compelling alternative to uniform quantization. By mapping weights or activations to a compact set of representative centroids, clustering mitigates quantization error and effectively handles outliers, as extreme values can be absorbed into centroids rather than expanding the overall dynamic range. Beyond its algorithmic robustness, clustering is also hardware-efficient: lookup table (LUT) implementations enable rapid centroid mapping and streamlined memory access, making it well suited for large-scale deployment. Notably, several commercial accelerators have already adopted such designs, including Apple's Neural Engine (Inc., 2024a) and Arm Ethos-U (Inc., 2020). The sparsity indexing mechanism in the Cerebras Wafer-Scale Engine (Inc., 2024b) further enables high-performance LUT implementation. Collectively, these developments underscore clustering as a practical, hardware-aligned solution for LUT-driven quantization.

Motivated by the challenge of activation outliers and the efficiency potential of LUTs, we present *CodeQuant*, a unified codebook-based clustering and quantization framework for low precision MoE models. CodeQuant exploits clustering robustness and LUT-based quantization to improve performance without any runtime overhead. Our contribution can be summarized as follows:

- We first introduce *Activation-oriented Outlier Smoothing* (AOS), which suppresses activation outliers through rotation matrix adjustment, effectively relocating them into the weight space.

- We then propose *Adaptive Weight Clustering with Centroid Finetuning* (ACCF) and *Permutation Invariant Outlier Grouping* (POG), which substantially reduce weight quantization error even in the presence of significant outliers.

- We develop a LUT kernel to demonstrate improvements in hardware efficiency. Across Phi-Mini-MoE-Instruct, Qwen3-30B-A3B, DeepSeek-V2-Lite and Mixtral 8x7B, CodeQuant consistently accelerates inference, lowers memory footprint, and preserves accuracy.

## 2 BACKGROUND AND RELATED WORK

### 2.1 OUTLIER IN LLMS

Activation outliers have been widely recognized as a major obstacle to effective quantization of LLMs since they expand the dynamic range and induce severe activation quantization errors. Prior work (Dettmers et al., 2022; Sun et al., 2024; An et al., 2025) highlights two predominant forms:

channel-wise outliers and massive activations. Moreover, residual connections exacerbate the problem by propagating outliers across layers and amplifying the adverse effects (Guo et al., 2024).

Mixture-of-Experts (MoE) LLMs are likewise affected by the outlier problem. Prior studies on MoE (Sun et al., 2024; Lo et al., 2025) report that massive activations frequently arise in the hidden states between decoder layers and are further propagated through residual connections, compounding their impact across subsequent layers. More recently, the notion of super experts has been introduced (Su et al., 2025), revealing an additional source of large-magnitude outliers specific to MoE architectures.

## 2.2 Outlier Aware Quantization

Prior efforts on LLM quantization have pursued two directions for addressing the outlier problem. The first explicitly isolates outliers and applies mixed-precision quantization (Dettmers et al., 2022; Kim et al., 2024; van Baalen et al., 2025; Huang et al., 2025; Dong & Zhang, 2025; Liu & Zhang, 2024), ensuring that extreme values are preserved at higher precision. The second seeks to mitigate outliers through invariant matrix transformations. Within this line, one strategy redistributes outliers between activations and weights (Xiao et al., 2024; Lin et al., 2024b; Xiang & Zhang, 2024). SmoothQuant (Xiao et al., 2024) is a representative work, which jointly smooths activations and weights to mitigate their impact. The other strategy is to smooth activation via orthogonal transformation. QuIP (Chee et al., 2024) and QuIP# (Tseng et al., 2024) initiate this line of work by leveraging rotation transformations to mitigate outliers. Building on this idea, QuaRot (Ashkboos et al., 2024) applies rotation to activations for outlier-free inference. DuQuant (Lin et al., 2024a) combines permutations for dual handling of outliers. SpinQuant (Liu et al., 2025) introduces learnable orthogonal rotation matrices that are optimized during post-training quantization, and subsequent work such as OSTQuant (Hu et al., 2025b) further incorporates a KL-based objective to fine-tune these rotations together with smoothing parameters. QSVD Wang et al. (2025) combines low-rank decomposition with rotation-based quantization, achieving superior accuracy and improved hardware efficiency.

In the context of weight quantization, most existing works nonetheless adopt uniform quantization schemes such as GPTQ (Frantar et al., 2022) and AWQ (Lin et al., 2024b), even though weight distributions in practice are far from uniform. To address this mismatch, early studies (Dettmers et al., 2023; Yoshida, 2023; Blumenberg et al., 2025) introduce quantile-based non-uniform quantization, leveraging the normal distributions assumption of weights to construct information-optimal codebooks. Meanwhile, SqueezeLLM (Kim et al., 2024) demonstrates that dynamic non-uniform quantization better adapts to the empirical weight distribution in LLMs. Building on earlier clustering-based compression techniques (Han et al., 2016; Xu et al., 2018), SqueezeLLM integrates K-means clustering into LLM quantization, yielding more robust results. Moreover, efficient algorithms for low-precision MoE remain largely underexplored. MoEQuant (Hu et al., 2025a) demonstrates that directly applying conventional quantization methods to MoE models yields suboptimal results, underscoring the importance of accounting for token–expert affinities.

## 2.3 LUT and Hardware Implementation

General Matrix Multiply (GEMM) with clustered multiplicands requires LUT support for efficient deployment. Without hardware-friendly LUTs, centroids must be stored as floating-point values and reloaded during computation, incurring significant overhead. Studies on both CPUs and GPUs address this by exploring LUT-based execution to bridge non-uniform quantization and practical deployment. On CPUs, DeepGEMM (Ganji et al., 2023) uses LUT-driven kernels for ultra-low-precision CNNs, LUTIN (Lin et al., 2024c) optimizes memory use via hyperparameter tuning, and T-MAC (Wei et al., 2025) reformulates mixed-precision GEMM as table lookup for LLM inference. On GPUs, LUT-GEMM (Park et al., 2024) and FLUTE (Guo et al., 2025) design optimized kernels to minimize unpacking overhead, while LUT Tensor Core (Mo et al., 2025) integrates LUT primitives into tensor-core pipelines through software–hardware co-design.

## 3 Methodology

The overview of CodeQuant is shown in Figure 1, which comprises three stages. In the first stage, we apply Activation-Oriented Outlier Smoothing (AOS) exclusively to the input activations, effectively mitigating activation outliers (Section 3.1). In the second stage, we optionally employ

Permutation-Invariant Outlier Grouping (POG), which reorders the columns of the weight matrix to better support the subsequent clustering process (Section 3.3). Stage three introduces Adaptive Weight Clustering and Centroid Finetuning (ACCF), which identifies optimal groupings and refines centroids to minimize output difference (Section 3.2). Finally, the resulting MoE is deployed using a LUT-based system, achieving superior computational efficiency (Section 3.4).

## 3.1 ACTIVATION-ORIENTED OUTLIER SMOOTHING

As illustrated in Figure 2, the rotational method introduces an additional matrix $R \in \mathbb{R}^{d_{in} \times d_{in}}$ applied to the activation $X \in \mathbb{R}^{N \times d_{in}}$ in both the Self-Attention (SA) and Feed-Forward Network (FFN). The SA blocks in MoE models share the same structure as those in standard LLMs, and therefore rotation transformation is invariant as discussed in (Ashkboos et al., 2024). The incorporation of rotational matrix $R$ within the FFN layers is illustrated in Figure 2. Although an FFN in MoE consists of a router and multiple experts, the router is simply a linear layer and each expert is structurally identical to a standard FFN. As a result, the MoE module

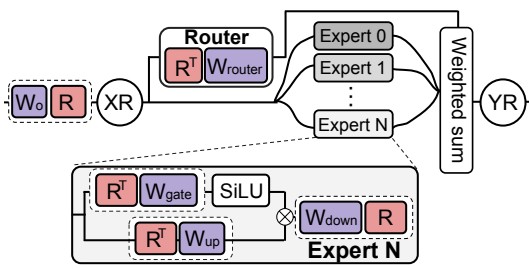

Figure 2: FFN layers within MoE is applied with rotational matrices for outlier smoothing.

is invariant to rotation transformations. To avoid introducing any online computation, we adopt an SA-style design in which the router and all experts share the same rotation matrix. Taking a single expert as an example, this rotational invariance can be expressed as follows:

$$(\phi(X_t RR^\top W_{gate}) \odot X_t RR^\top W_{up})W_{down} = (\phi(XW_{gate}) \odot XW_{up})W_{down} \tag{1}$$

where $\phi(\cdot)$ denotes a nonlinear activation function (e.g., SiLU) and $X_t \in \mathbb{R}^{n \times d_{in}}$ denotes the subset of tokens assigned to that expert.

Although random rotation improves quantization, replacing weight quantization with clustering reveals activation quantization as the dominant bottleneck. To address this, we fine-tune the rotation matrix $R$ via the Cayley transform to smooth activations (Nishimori & Akaho, 2005; Li et al., 2020). Specifically, for any matrix $M \in \mathbb{R}^{d_{in} \times d_{in}}$, where $d_{in}$ denotes the model's hidden dimension, we first extract its skew-symmetric component and then derive an orthogonal matrix via the Cayley transform:

$$S = \tfrac{1}{2}(M - M^\top) \qquad R = (I - S)(I + S)^{-1} \tag{2}$$

By parameterizing the matrix $M$, this construction guarantees that the matrix $R$ remains orthogonal while keeping the process fully differentiable. AOS employs learnable rotation matrices to minimize the quantization error of rotated activations, defined as $X_R = XR$. By minimizing the quantization error of rotated activations, the rotation explicitly reduces the influence of outliers on the activation side, leaving the weights to accommodate more of the variation. Formally, the optimization objective is defined as:

$$\arg\min_R ||X_R - Q(X_R)||^2 \tag{3}$$

where $Q(\cdot)$ denotes the quantization function (i.e. integer quantization). Using WikiText2 (Merity et al., 2016) as the calibration dataset, we observe a consistent reduction in quantization error during fine-tuning. On the held-out test set, fine-tuned rotations yield lower quantization error than random rotations, demonstrating that the learned rotations generalize beyond calibration. We provide ablation study on fine-tuned rotation in Section 4.4.

## 3.2 ADAPTIVE WEIGHT CLUSTERING AND CENTROID FINETUNING

Building on the smoothed input activations enabled by AOS, we introduce the ACCF method, which refines grouping and centroid search to further reduce clustering error in the outputs of matrix products. Specifically, let $W_R = R^\top W$ denote rotated weight matrix. We adopt a row-wise parameterization for clustering. Specifically, let $C \in \mathbb{R}^{d_{out} \times K}$ denote the centroid matrix where the $i$-th row $C_{i,:}$ serves as the codebook for the $i$-th row of weights, and let $A \in \{0, 1\}^{d_{out} \times d_{in} \times K}$ be the binary assignment

tensor satisfying $\sum_{k=1}^{K} A_{i,j,k} = 1$. The reconstructed weight matrix $W_c$ is defined element-wise as:

$$W_{c;ij} = \sum_{k=1}^{K} C_{i,k} A_{i,j,k} \tag{4}$$

To minimize the changes in the output, we set the target as:

$$\arg\min_{C,A} ||X_R W_R - \tilde{X}_R W_c||^2 \tag{5}$$

where $\tilde{X}_R \in \mathbb{R}^{N \times d_{in}}$ denotes the input activations at this layer when the upstream weights have already been clustered. Equation 5 specifies the objective function for enabling matrix computations within the SA layers of the MoE through the hybrid operation of input quantization and weight clustering.

However, unlike in SA, applying the same operation to the routing mechanism of the MoE FFN may cause mismatches in token-expert assignments compared with the original MoE, thereby degrading performance. To address this, we design a MoE-specific objective utilizing MoE weighted sum. Meanwhile, prior works have shown the importance of token–expert affinity (Dai et al., 2022; Li et al., 2025; Hu et al., 2025a; Liang et al., 2025). Thus, we add a KL divergence loss on router logits during fine-tuning to preserve the original token–expert assignment. In general, we modify the objective function in Equation 5 as follows:

$$\mathcal{L} = \begin{cases} ||X_R W_R - \tilde{X}_R W_c||^2, & \text{if } W_R \in \{W_{R;Q}, W_{R;K}, W_{R;V}\}, \\ ||Y - \sum_{i=1}^{E} \tilde{\Pi}_i \tilde{X}_R W_c||^2 + \lambda D_{\mathrm{KL}}(\tilde{\Pi}, \Pi), & \text{if } W_R \in \{W_{R;gate}, W_{R;up}\}, \end{cases} \tag{6}$$

where $E$ denotes the number of experts, $Y \in \mathbb{R}^{N \times d_{in}}$ is the weighted sum produced by the MoE module on the calibration set using the non-clustered weights, and $\tilde{\Pi}$ and $\Pi$ represent the router outputs corresponding to $\tilde{X}_R$ and $X_R$, respectively. $D_{KL}(\cdot, \cdot)$ returns the KL divergence between the two inputs and $\lambda$ specifies the relative importance of the objective functions.

The optimization problems in Equation 6 can be addressed in an alternating, iterative manner. We first fix the assignment matrix $A$ and optimize the centroid matrix $C$. To this end, we employ a local fine-tuning procedure following Equation 6 to update $C$ via gradient descent. To determine the assignment matrix $A$ given the centroids $C$ while minimizing the output difference, a straightforward approach is to use the nearest-neighbor rounding method as in the standard K-means algorithm. However, this does not perfectly align with the objective functions in Equation 6. To mitigate this, we design an analytical solution derived using gradient to update assignment matrix. For ease of illustration, we adopt the loss function defined in Equation 5, though a similar technique can also be applied to the loss function in Equation 6. We begin by deriving the gradient expression for the clustered weights.

$$\nabla_{W_c} = \frac{\partial \mathcal{L}}{\partial W_c} = 2\tilde{X}_R^\top \tilde{X}_R W_c - 2\tilde{X}_R^\top X_R W_R \tag{7}$$

Set $\hat{D}_1 = \tilde{X}_R^\top \tilde{X}_R$ and $\hat{D}_2 = \tilde{X}_R^\top X_R$. For computational efficiency, we approximate these matrices by retaining only their diagonal entries, i.e., $D_1 = \mathrm{Diag}(\hat{D}_1)$, $D_2 = \mathrm{Diag}(\hat{D}_2)$. For each element $W_{R;ij}$, the corresponding error introduced by assigning it to the $k$-th centroid $C_{i,k}$ is:

$$\psi(W_{R,ij}, C_{i,k}) = ||D_{1,jj}, C_{i,k} - D_{2,jj}, W_{R,ij}||^2 \tag{8}$$

where $D_{1,jj}$ and $D_{2,jj}$ denote the $j$-th diagonal elements of $D_1$ and $D_2$, respectively. Hence, the optimal assignment for $W_{c,ij}$ is obtained by searching over the row-specific centroids $\{C_{i,k}\}_{k=1}^{K}$:

$$k^* = \arg\min_{k \in \{1,...,K\}} \psi(W_{R,i,j}, C_{i,k}) \tag{9}$$

### 3.3 PERMUTATION-INVARIANT OUTLIER GROUPING

The ACCF algorithm described in Section 3.2 is directly applied to the rotated weight matrices $W_R$. In practice, achieving strong MoE accuracy under ACCF critically depends on initializing $W_R$ to be cluster-friendly, such that a low-error clustered solution can be readily obtained. Since AOS minimizes the quantization error of rotated activations only, the remaining variability is left to the weights, making a cluster-friendly initialization crucial for ACCF to achieve high performance.

However, in practice, we observe that $W_R$ is sometimes not amenable to clustering, as shown in Figure 3 (a). Consider a weight vector $W_R$ partitioned into clustering groups of size $g = 4$, highlighted by the orange boxes. Each clustering group is allocated a centroid budget of $k = 2$. Owing to the high variance within group 1, the optimal clustering solution still incurs a clustering error of 17. To reduce the error, we propose the POG method, as illustrated in Figure 3 (c). Specifically, the weight vector is first divided into smaller sub-groups (shown in green boxes),

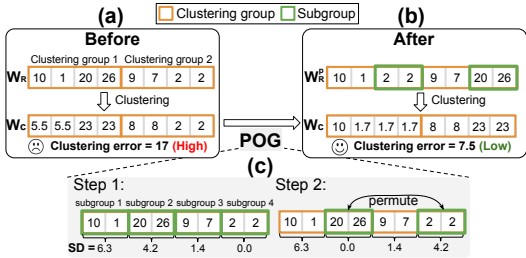

Figure 3: The overview of the POG framework.

each of size 2 in this example. In Step 1, the variance is computed across the elements within each sub-group. In Step 2, the sub-groups are permuted as indivisible units, ordered by their variance, so as to redistribute high- and low-variance sub-groups more evenly across the larger groups of size $g = 4$. This reordering helps reduce the variance within each clustering group and thereby lowers the overall clustering error for the resultant $W_R^p$, as shown in Figure 3 (b). The key intuition is that, in the original $W_R$, group 1 contains weights that would require more than two centroids to achieve low error, while group 2 is much easier to cluster. By permuting elements at the sub-group level, we obtain a more cluster-friendly $W_R^p$. It is important to note that this idea differs from prior work designed to facilitate quantization (Lin et al., 2024a), since the reordered matrix $W_R^p$ is not necessarily amenable to quantization. The resultant $W_R^p$ is then used as the initialization for the subsequent ACCF operations, and leading to improved performance. The detailed POG algorithm is shown in the Appendix A.2.1.

In practice, directly using the permuted matrix $W_R^p$ alters the output will lead to incorrect results. Prior work (Lin et al., 2024a) addresses this by formulating permutation as a matrix multiplication. Specifically, permuting $W_R$ can be achieved by multiplying it with a permutation matrix $P$, which encodes the permutation pattern shown in Figure 3. As an orthogonal matrix, $P$ can be folded into the SA and FFN components of MoE using the same method as rotation matrix $R$. In CodeQuant, the permutation matrices $P$ and $P^\top$ are introduced after $W_v P$ and $P^\top W_{out}$ in the self-attention block, and after $W_{up} P$ as well as before $P^\top W_{down}$ in the feed-forward block, ensuring output invariance and improving ACCF performance.

## 3.4 CODEQUANT KERNEL AND SYSTEM IMPLEMENTATION

To evaluate the potential real-world performance of CodeQuant, we design and simulate an efficient LUT-based GEMM kernel. While a full hardware implementation is beyond the scope of this work, our simulation, based on the validated Accel-Sim framework (Mo et al., 2025; Guo et al., 2023; Avalos Baddouh et al., 2021), models realistic architectural modifications. First, the input and weight matrices are tiled by the weight group size. Each group of weights shares the same set of centroids and is multiplied with multiple activation channels, as shown in Figure 4 (a). To reduce redundant multiplications, for each weight group we precompute a LUT using the 16 centroid values and the 16 possible 4-bit integer activation values, as shown in step 1 of Figure 4 (b). The LUT consists of 16 subtables, each computed from one centroid value over 16 activation values when the activations are quantized to 4-bit. CodeQuant uses a two-level Mux to select the output as shown in step 2 in Figure 4 (b). By pairing activation and weight for shared-memory access, shared-memory conflicts are reduced compared with separate activation and weight accesses (Guo et al., 2025). The LUT resides in SM shared memory, as shown in Figure 4 (c) and occupies only a small fraction of the shared memory available on modern GPUs (NVIDIA Corporation, a;b).

Although CodeQuant GEMM kernel is promising due to its advantages in eliminating dequantization and multiplication through simple table lookup, existing GPU implementation still faces challenges. This is mainly due to limited instruction support for efficient lookup table precomputation (Mo et al., 2025) and shared memory bank conflicts from extensive random indexing operations (Guo et al., 2025). To make better use of the precomputed lookup tables, the number of activation channels in the input matrix in Figure 4 (a) should increase. However, modern GPU uses the CUDA tensor core for high performance matrix multiplication and the tensor core instruction only supports a fixed size of matrix tiles multiplication ($8 \times 4 \times 16$ INT8 matrix multiplication in Nvidia RTX A100 GPU (NVIDIA Corporation, a)). To achieve better LUT-based GEMM performance and keep a fair

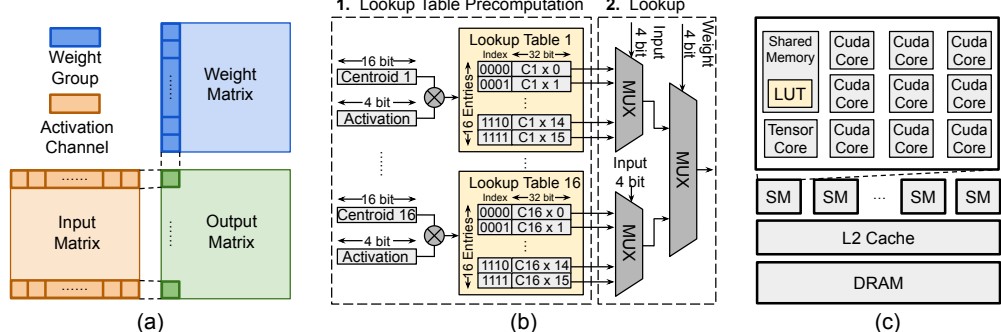

Figure 4: (a) One tile of the matrix multiplication. (b) The steps of CodeQuant kernel, including a one-time lookup table precomputation and table lookup. (c) The precomputed lookup table is stored in the shared memory in the Streaming Multiprocessors (SM) in GPU.

comparison with tensor cores, we simulate the GPU performance with optimized matrix sub-tile shape under the same floating point operation numbers per cycle using Accel-Sim (Khairy et al., 2020). To mitigate the bank conflicts, the LUT can be duplicated into more memory banks (Lo et al., 2025) to reduce the chance of multiple threads accessing the same memory bank. To keep the same total shared memory size, we can increase the number of banks (32 banks in A100 GPU) and reduce the size of each memory bank, which requires the shared memory structure improvement. We use Accel-Sim to simulate the LUT-based GEMM performance with optimized GPU shared memory structure.

## 4 EXPERIMENTS

We evaluate CodeQuant across MoE models of varying sizes and architectures, including Phi-mini-MoE-Instruct (Abdin et al., 2024), Qwen3-30B-A3B (Yang et al., 2025), DeepSeek-V2-Lite (DeepSeek-AI et al., 2024), and Mixtral 8x7B (Jiang et al., 2024). The evaluations cover language generation, commonsense QA tasks, and mathematical reasoning tasks. For language modeling, we report perplexity on WikiText2 (Merity et al., 2016) and C4 (Raffel et al., 2023). For zero-shot QA, we measure accuracy on ARC (Clark et al., 2018), HellaSwag (Zellers et al., 2019), MMLU (Hendrycks et al., 2021), PIQA (Bisk et al., 2020), and WinoGrande (Sakaguchi et al., 2021). For few-shot mathematical reasoning, we evaluate CodeQuant using GSM8K (8-shot) (Cobbe et al., 2021) and MATH500 (4-shot) (Hendrycks et al., 2021).

In the AOS stage, we apply the Cayley transform to optimize the activation-quantization rotation matrix $R$, using 1,024 WikiText2 samples over 128 iterations. In the ACCF stage, we optimize centroids over 64 iterations with 512 WikiText2 calibration samples, setting the KL divergence coefficient $\lambda$ to 1.0. We study the impact of $\lambda$ in Section 4.4. In terms of preprocessing time, the AOS stage requires approximately 15/20/30/50 minutes for Phi-mini-MoE-Instruct, DeepSeek-V2-Lite, Qwen3-30B-A3B and Mixtral 8x7B on H100 GPUs, respectively. The subsequent ACCF stage requires 30/40/110/240 minutes for the same models. Despite the preprocessing cost, our framework is fully offline meaning no on-the-fly computation. During inference, the weight matrices remain fixed, and inference proceeds in the same way as a standard MoE. Section **??** shows that this leads to a net inference speedup.

We compare CodeQuant with several PTQ methods, including RTN (Round-to-Nearest), SmoothQuant (Xiao et al., 2024), QuaRot (Ashkboos et al., 2024), SqueezeLLM (Kim et al., 2024), DuQuant (Lin et al., 2024a) and SpinQuant (Liu et al., 2025) as baseline methods. For methods that rely on online Hadamard transforms, we adopt the same setting to ensure methodological consistency.

We use the same activation bitwidth across methods, including SqueezeLLM, where input activations are quantized with RTN. For weights, we match the total number of discrete representation values. For instance, when QuaRot uses 4-bit quantization, we configure CodeQuant with 16 centroids for weight clustering to yield an equivalent representation capacity, using the same centroid-selection strategy as SqueezeLLM. All algorithms are evaluated under two quantization/clustering configurations. In

Table 1: Performance in perplexity (PPL) on Wiki2 and C4 dataset, and accuracy on Arc-Challenge (A-c), Arc-easy (A-e), HellaSwag (HS), MMLU (ML), PIQA (PQ) and WinoGrande (WG). For each setting, we report the BF16 baseline in the first row. More results are shown in the Appendix.

| | Models | Methods | Wiki2 ($\downarrow$) | C4 ($\downarrow$) | A-c ($\uparrow$) | A-e ($\uparrow$) | HS ($\uparrow$) | ML ($\uparrow$) | PQ ($\uparrow$) | WG ($\uparrow$) | Avg ($\uparrow$) |
|---|---|---|---|---|---|---|---|---|---|---|---|
| A4W4 Embedding-wise | Phi-mini-MoE-Instruct | BF16 | 6.83 | 13.06 | 0.581 | 0.813 | 0.759 | 0.681 | 0.797 | 0.753 | 0.731 |
| | | RTN | 9811.22 | 7431.27 | 0.287 | 0.268 | 0.261 | 0.232 | 0.501 | 0.516 | 0.344 |
| | | SqueezeLLM | 8383.63 | 5619.01 | 0.279 | 0.281 | 0.263 | 0.236 | 0.515 | 0.500 | 0.346 |
| | | SmoothQuant | 24071.25 | 16320.79 | 0.263 | 0.280 | 0.270 | 0.240 | 0.528 | 0.503 | 0.347 |
| | | QuaRot | 7.93 | 14.44 | **0.545** | 0.784 | 0.725 | 0.633 | 0.775 | 0.702 | 0.694 |
| | | **CodeQuant** | **7.63** | **13.94** | 0.538 | **0.790** | **0.728** | **0.644** | **0.784** | **0.716** | **0.700** |
| | DeepSeek-V2-Lite | BF16 | 6.69 | 9.32 | 0.491 | 0.759 | 0.780 | 0.551 | 0.804 | 0.709 | 0.682 |
| | | RTN | 812.90 | 660.45 | 0.226 | 0.295 | 0.283 | 0.237 | 0.513 | 0.483 | 0.339 |
| | | SqueezeLLM | 806.71 | 614.70 | 0.257 | 0.301 | 0.277 | 0.238 | 0.541 | 0.508 | 0.354 |
| | | SmoothQuant | 11.57 | 16.10 | 0.381 | 0.645 | 0.658 | 0.305 | 0.747 | 0.581 | 0.553 |
| | | QuaRot | 7.75 | 10.75 | 0.457 | 0.720 | 0.745 | 0.450 | 0.787 | 0.682 | 0.640 |
| | | **CodeQuant** | **7.08** | **9.85** | **0.479** | **0.749** | **0.767** | **0.515** | **0.791** | **0.684** | **0.664** |
| | Qwen3-30B-A3B | BF16 | 9.04 | 14.05 | 0.566 | 0.793 | 0.776 | 0.778 | 0.805 | 0.694 | 0.735 |
| | | RTN | 181.59 | 232.49 | 0.230 | 0.385 | 0.367 | 0.236 | 0.565 | 0.445 | 0.371 |
| | | SqueezeLLM | 100.47 | 121.55 | 0.222 | 0.352 | 0.367 | 0.243 | 0.576 | 0.504 | 0.377 |
| | | SmoothQuant | 23.01 | 33.39 | 0.383 | 0.584 | 0.490 | 0.413 | 0.717 | 0.547 | 0.522 |
| | | QuaRot | 16.04 | 24.27 | 0.386 | 0.596 | 0.609 | 0.585 | 0.735 | 0.575 | 0.581 |
| | | **CodeQuant** | **10.31** | **15.75** | **0.522** | **0.757** | **0.688** | **0.735** | **0.780** | **0.685** | **0.694** |
| | Mixtral-8x7B | BF16 | 4.01 | 7.41 | 0.579 | 0.851 | 0.720 | 0.677 | 0.856 | 0.799 | 0.747 |
| | | RTN | 10502.14 | 14045.38 | 0.319 | 0.261 | 0.284 | 0.243 | 0.492 | 0.504 | 0.350 |
| | | SqueezeLLM | 13952.66 | 19725.12 | 0.297 | 0.282 | 0.279 | 0.251 | 0.527 | 0.519 | 0.359 |
| | | SmoothQuant | 77.32 | 96.01 | 0.222 | 0.349 | 0.303 | 0.236 | 0.565 | 0.497 | 0.362 |
| | | QuaRot | 16.79 | 24.29 | 0.348 | 0.570 | 0.512 | 0.286 | 0.708 | 0.560 | 0.497 |
| | | **CodeQuant** | **4.65** | **8.06** | **0.565** | **0.819** | **0.715** | **0.644** | **0.827** | **0.780** | **0.725** |
| A4W4 Block-wise | Phi-mini-MoE-Instruct | RTN | 20.86 | 30.75 | 0.345 | 0.540 | 0.475 | 0.318 | 0.657 | 0.529 | 0.477 |
| | | SqueezeLLM | 12.44 | 20.21 | 0.399 | 0.607 | 0.590 | 0.455 | 0.687 | 0.572 | 0.552 |
| | | SmoothQuant | 15.34 | 24.18 | 0.356 | 0.559 | 0.532 | 0.464 | 0.656 | 0.577 | 0.524 |
| | | QuaRot | 7.63 | 13.82 | 0.534 | 0.790 | 0.728 | 0.633 | 0.783 | 0.719 | 0.698 |
| | | **CodeQuant** | **7.28** | **13.54** | **0.562** | **0.800** | **0.733** | **0.646** | **0.792** | **0.729** | **0.710** |
| | DeepSeek-V2-Lite | RTN | 161.08 | 159.65 | 0.236 | 0.368 | 0.344 | 0.236 | 0.581 | 0.515 | 0.380 |
| | | SqueezeLLM | 115.66 | 112.59 | 0.238 | 0.379 | 0.364 | 0.234 | 0.590 | 0.500 | 0.384 |
| | | SmoothQuant | 9.11 | 12.72 | 0.387 | 0.652 | 0.687 | 0.347 | 0.761 | 0.613 | 0.574 |
| | | QuaRot | 7.62 | 10.59 | 0.462 | 0.719 | 0.745 | 0.483 | 0.781 | 0.668 | 0.643 |
| | | **CodeQuant** | **7.03** | **9.79** | **0.480** | **0.741** | **0.764** | **0.525** | **0.794** | **0.698** | **0.667** |

the first, referred to as **Block-wise**, quantization or clustering is applied within groups of $g = 1024$ weight values along the embedding dimension. In the second, termed **Embedding-wise**, quantization is applied across the entire embedding dimension, spanning the full embedding vector.

We evaluate CodeQuant GEMM kernel using Accel-Sim (Khairy et al., 2020), a state-of-the-art GPU simulator, configured to model an A100 80GB GPU with CodeQuant-optimized tensor cores. Detailed simulation settings are provided in the Appendix A.3. As baselines on real A100 hardware, we measure the latencies of HuggingFace (Wolf et al., 2020) BF16 models, QuaRot (Ashkboos et al., 2024) A4W4 quantized models, and SqueezeLLM (Kim et al., 2024) A4W4 quantized models. SqueezeLLM serves as a baseline for weight clustering and activation quantization without GPU architectural modification, helping isolate the latency performance gains from CodeQuant hardware kernel design. Experiments use a prefill length of 512, decoding length of 128, and batch size of 16. Additionally, we measure the real hardware performance of CodeQuant by benchmarking the A8W4 T-MAC kernel (Wei et al., 2025), a mixed-precision LUT-based CPU GEMM kernel, against Llama.cpp (Gerganov & ggml-org contributors, 2023) BF16 and A8W4 models on CPU.

## 4.1 MAIN RESULTS

Table 1 summarizes the evaluation results of CodeQuant under different configurations. For clarity, we adopt the 'AxWx' notation. For instance, in QuaRot, RTN, and SmoothQuant, 'A4W4' denotes 4-bit quantization of activations and 4-bit quantization of weights. In contrast, under CodeQuant, 'A4W4' corresponds to applying 4-bit linear quantization to activations and clustering weights into $2^4 = 16$ centroids. In the Embedding-wise setting, POG has no effect on the final performance. Therefore, POG is not applied here. Detailed explanation will be provided in Appendix A.2.2.

We first present the Embedding-wise evaluation results. For A4W4, CodeQuant delivers substantial improvements over existing methods. On Qwen3-30B-A3B, it reduces perplexity by 5.73 on

Table 2: Rotation-based method performance comparison. CodeQuant$_{had}$ indicates that online Hadamard transforms are enabled during the quantization process.

| | Models | Methods | Wiki2 (↓) | C4 (↓) | A-c (↑) | A-e (↑) | HS (↑) | ML (↑) | PQ (↑) | WG (↑) | Avg (↑) |
|---|---|---|---|---|---|---|---|---|---|---|---|
| A4W4 Embedding-wise | DeepSeek-V2-Lite | DuQuant | 8.43 | 11.94 | **0.455** | 0.708 | 0.623 | 0.400 | 0.775 | **0.693** | 0.624 |
| | | SpinQuant$_{had}$ | 9.24 | 12.71 | 0.427 | 0.692 | 0.706 | 0.425 | 0.774 | 0.638 | 0.610 |
| | | **CodeQuant$_{had}$** | **8.16** | **11.38** | 0.445 | **0.723** | **0.727** | **0.454** | **0.782** | 0.644 | **0.629** |
| | Qwen3-30B-A3B | DuQuant | 13.52 | 20.10 | 0.472 | 0.662 | 0.687 | 0.654 | **0.739** | 0.606 | 0.637 |
| | | SpinQuant$_{had}$ | 14.61 | 22.07 | 0.415 | 0.600 | 0.628 | 0.584 | 0.692 | 0.622 | 0.590 |
| | | **CodeQuant$_{had}$** | **12.69** | **19.89** | **0.477** | **0.697** | **0.691** | **0.679** | **0.739** | **0.635** | **0.653** |

WikiText2 and 8.52 on C4, while increasing average accuracy by 11.3% compared to QuaRot, with even larger gains over SmoothQuant on both metrics. On DeepSeek-V2-Lite, CodeQuant again improves performance, lowering perplexity by 0.67 on WikiText2 and 0.9 on C4, alongside a 2.4% accuracy increase over QuaRot. On Mixtral 8×7B, CodeQuant shows the same trend, reducing perplexity by 12.14 on WikiText2 and 16.23 on C4 compared to QuaRot, and increasing average accuracy by 22.8%. These results highlight CodeQuant's consistent advantages across architectures and demonstrate that its effectiveness remains stable across both model structure and model scales. The A8W4 Embedding-wise results are detailed listed in Appendix A.4.

With Block-wise setting and POG enabled, we evaluate Phi-mini-MoE-Instruct and DeepSeek-V2-Lite. Under A4W4, both models show clear improvements over the Embedding-wise baseline. However, when moving to A8W4, Phi-mini-MoE-Instruct benefits only marginally, and DeepSeek-V2-Lite even drops by 0.3% relative to the baseline. We attribute this to DeepSeek's already strong accuracy without POG, with less than a 1% gap compared to BF16. These results suggest that permutation is effective under extreme compression, as detailed in Appendix A.4.

In addition, we evaluate CodeQuant against two strong rotation-based PTQ baselines, SpinQuant and DuQuant, both of which suppress outliers through trainable or structured transformations and utlize online Hadamard transformation. For fairness, we adopt online Hadamard transforms and denote this variant as CodeQuant$_{had}$, matching the SpinQuant$_{had}$ setup. As shown in Table 2, CodeQuant$_{had}$ consistently outperforms both baselines. On Qwen3-30B-A3B, it reaches an average accuracy of 0.653 compared to 0.637 for DuQuant and 0.590 for SpinQuant$_{had}$. On DeepSeek-V2-Lite, it achieves 0.629, again exceeding DuQuant at 0.624 and SpinQuant$_{had}$ at 0.610, demonstrating robust advantages across language modeling and downstream tasks.

## 4.2 MATHEMATICALLY REASONING PERFORMANCE

We further assess whether CodeQuant preserves reasoning-heavy capabilities, which are typically more sensitive to quantization. We evaluate DeepSeek-V2-Lite and Qwen3-30B-A3B under the A4W4 Embedding-wise configuration on GSM8K (8-shot) and MATH500 (4-shot) (DeepSeek-AI et al., 2024), where each $k$-shot prompt includes $k$ worked examples before the test question. As shown in Table 3, CodeQuant substantially outperforms QuaRot and remains close to the BF16 baseline. On DeepSeek-V2-Lite, the degradation is minimal, only 3.4% on

Table 3: A4W4 Embedding-wise results on GSM8K (8-shot) and MATH500 (4-shot).

| Models | Methods | GSM8K (↑) | MATH500 (↑) |
|---|---|---|---|
| DeepSeek-V2-Lite | BF16 | 0.364 | 0.121 |
| | QuaRot | 0.231 | 0.093 |
| | **CodeQuant** | **0.330** | **0.108** |
| Qwen3-30B-A3B | BF16 | 0.924 | 0.322 |
| | QuaRot | 0.508 | 0.128 |
| | **CodeQuant** | **0.867** | **0.241** |

GSM8K and 1.3% on MATH500. For Qwen3-30B-A3B, the advantage becomes even more pronounced: CodeQuant improves over QuaRot by 35.9% on GSM8K, and 11.3% on MATH500, highlighting its strength on reasoning-heavy tasks.

## 4.3 LATENCY EVALUATION

Figure 5 presents the normalized speedups of all baselines, with BF16 latency normalized to 1. Compared with the BF16 models, CodeQuant achieves an average 2.63× speedup, which underscores the effectiveness of low-bit activation and weight quantization together with the LUT-based GEMM

Table 4: AOS Impact

| Method | DeepSeek-V2-Lite | |
|---|---|---|
| Random | Wiki2 ↓ | 7.29 |
| | C4 ↓ | 10.16 |
| | Acc ↑ | 0.652 |
| **AOS** | Wiki2 ↓ | **7.06** |
| | C4 ↓ | **9.85** |
| | Acc ↑ | **0.667** |

Table 5: KL Loss Impact

| Method | Task | Phi-mini | Deepseek-V2-Lite |
|---|---|---|---|
| W/O KL | Wiki2 ↓ | 7.29 | 7.10 |
| | C4 ↓ | 13.95 | 9.87 |
| | Acc ↑ | 0.694 | 0.658 |
| **W/ KL** | Wiki2 ↓ | **7.06** | **7.03** |
| | C4 ↓ | **13.80** | **9.79** |
| | Acc ↑ | **0.700** | **0.667** |

Table 6: Bitwidth Budgets Impact

| Model | DeepSeek-V2-Lite | | |
|---|---|---|---|
| | Wiki2 ↓ | C4 ↓ | Acc ↑ |
| SqueezeLLM$_{A4W2}$ | 24.36 | 32.98 | 0.496 |
| SqueezeLLM$_{A4W3}$ | 8.40 | 11.69 | 0.619 |
| SqueezeLLM$_{A4W4}$ | 7.17 | 10.01 | 0.652 |
| **CodeQuant**$_{A4W2}$ | 10.68 | 14.59 | 0.568 |
| **CodeQuant**$_{A4W3}$ | 7.59 | 10.58 | 0.639 |
| **CodeQuant**$_{A4W4}$ | 7.06 | 9.85 | 0.667 |

design. The speedup of CodeQuant over QuaRot highlights the advantage of replacing repetitive multiply-accumulate operations with direct LUT indexing, thereby reducing redundant multiplications. The improvement over SqueezeLLM reflects the benefit of deploying a GPU implementation that uses optimized LUT operations. Considering the strong accuracy results of CodeQuant shown in Table 1, CodeQuant achieves the optimal performance among the baselines. Furthermore, we validate these performance trends on real hardware by benchmarking a CPU kernel, where CodeQuant achieves up to $4.15\times$ speedup over a BF16 baseline. Additional evaluations can be found in Appendix A.5.

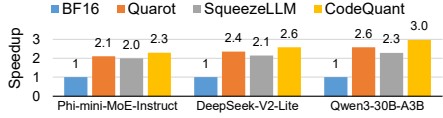

Figure 5: Normalized speedup on one A100 GPU.

## 4.4 Ablation Studies

**Impact of Activation Smoothing** We evaluate whether fine-tuning the rotation matrix improves accuracy on DeepSeek-V2-Lite under the A4W4 Embedding-wise configuration, keeping all other settings fixed. Specifically, we compare a random rotation with its fine-tuned version produced by AOS. As shown in Table 4, rotational matrix finetuning yields consistent improvements, boosting accuracy by 1.4% and reducing perplexity by 0.23 on WikiText2 and by 0.31 on C4.

**Impact of KL Penalty** We evaluate the effectiveness of the KL divergence term defined in Equation 6. The ablation is conducted on Phi-mini-MoE-Instruct and DeepSeek-V2-Lite under the A4W4 Block-wise configuration, comparing two settings: (i) centroids fine-tuned without the KL divergence term ($\lambda = 0.0$), and (ii) centroids optimized with the full ACCF loss ($\lambda = 1.0$). As shown in Table 5, ACCF with the KL penalty outperforms the version without it. Additional analysis in Appendix A.7 further shows that the KL penalty also stabilizes the router behavior, indicating that KL regularization helps preserve the original expert-routing pattern after quantization.

**Impact of Low-Bit Compression** We examine CodeQuant performance under different centroid budgets on DeepSeek-V2-Lite with Embedding-wise quantization. We apply the same rotation matrix to quantize activations for both CodeQuant and SqueezeLLM, and evaluate over three settings: A4W2, A4W3, and A4W4. As shown in Table 6, CodeQuant consistently outperforms SqueezeLLM across all budgets. Under the most aggressive case (A4W2), CodeQuant's average accuracy decreases by 9.9% relative to the A4W4 case, whereas SqueezeLLM drops by 15.6%. Moreover, CodeQuant's advantage widens as the budget shrinks from 1.5% at A4W4 to 7.2% at A4W2, indicating robustness under extreme compression.

## 5 Conclusion

We present CodeQuant, a unified quantization-and-clustering framework for low-precision MoE. CodeQuant reduces quantization error while preserving accuracy, achieves up to $4.15\times$ latency reduction. Experiments confirm that CodeQuant delivers superior accuracy–efficiency trade-offs compared to other baseline algorithms, enabling more reliable low-precision deployment of MoE.

## ETHICS STATEMENT

This work complies with the ICLR Code of Ethics. CodeQuant is a post-training quantization framework evaluated on pretrained models and public datasets, without the use of private or user-specific data. Our research does not involve human subjects, private or sensitive data, or personally identifiable information. The method modifies only internal representations through weight quantization and routing, introducing no new risks in fairness, privacy, or security beyond those inherent to the base models. We are not aware of any direct ethical concerns specific to this work.

## REPRODUCIBILITY STATEMENT

Code and models: All experiments in this paper are conducted on publicly available datasets with specified preprocessing steps. Detailed configurations, including hyperparameter, training procedures, and hardware specifications, are reported in the experiment section. Baselines are re-implemented following their original papers, with reference to the authors' released code when available. While the source code for CodeQuant is not released at submission time, we will make it publicly available upon acceptance to facilitate reproducibility.

Datasets: All datasets used in this work are publicly available.

Randomness: All experiments are run with fixed random seeds in the scripts, to ensure consistent results.

Compute resources: Our experiments are conducted on NVIDIA RTX H100, RTX A100, Accel-Sim GPU simulator, and Intel CPU as described in Section 4.

## USE OF LARGE LANGUAGE MODELS

Large language models (LLMs), such as ChatGPT, were used only for polishing language and improving readability. All technical ideas, analyses, experiments, and conclusions were conceived, implemented, and validated by the authors. The final manuscript was carefully reviewed to ensure accuracy and correctness.

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

## A  APPENDIX

### A.1  ROTATION MATRIX IN DEEPSEEK-V2-LITE

In Section 3.1, we integrate the rotation matrix into the weight parameters. Due to the architectural differences between the DeepSeek-V2-Lite model and Qwen3-30B-A3B, in DeepSeek-V2-Lite SA block, the rotation matrices are applied to $W_q$ and $W_{kv\_a}$. In the MoE FFN block, DeepSeek-V2-Lite includes a shared expert; therefore, the rotation matrices are also applied to the shared expert's $W_{up}$ and $W_{gate}$.

### A.2  POG ALGORITHM AND ANALYSIS

#### A.2.1  ALGORITHM

In Section 3.3 we propose a permutation method. In this section, we will introduce how to construct a permutation matrix $P$ that makes the weights more amenable to clustering in detail.

First, we need to obtain a permutation sequence using Algorithm 1. Given a weight matrix $W_R \in \mathbb{R}^{d_{in} \times d_{out}}$, we compute a permutation sequence $\pi \in \mathbb{R}^{d_{out}}$, defined as a bijective sequence in which each element $\pi_i$ specifies the original column relocated to the $i$-th position in the permuted

---

**Algorithm 1:** POG Algorithm

**Input:** $W_R \in \mathbb{R}^{d_{in} \times d_{out}}$ is the weight matrix after rotation; $g \in \mathbb{N}$ is the quantization group size; $g_s \in \mathbb{N}$ is the small subgroup size, which is the unit to swap, and it satisfies $g_s < g$.

**Output:** A column permutation order $\pi$ of $\{1, \ldots, d_{out}\}$.

1 **Procedure**

2    $N_g \leftarrow d_{out}/g, \quad N_s \leftarrow d_{out}/g_s, \quad n \leftarrow g/g_s$;

3    Compute the mean absolute value of each column: $S \in \mathbb{R}^{d_{out}}$ where $s_j = \frac{1}{d_{in}} \sum\limits_{r=1}^{d_{in}} |W_{R;rj}|$;

4    $I_{idx} \leftarrow \text{argsort}(S, \text{desc})$;

5    Partition $I_{idx}$ into $N_s$ groups of size $g_s$, such that each group
     $G_i = I_{idx}[(i-1)g_s + 1 : ig_s] \in \mathbb{R}^{g_s}, \quad i = 1, \ldots, N_s$;

6    **for** $i = 1$ **to** $N_s$ **do**

7      $W_{G_i} = W_R[:, G_i]$;

8      $v_i \leftarrow \text{Mean}(\text{StdDev}(W_{G_i}, \dim = 1), \dim = 0)$;

9    $V = \{v_1, \ldots, v_{N_s}\}$;

10    $\check{I}_V \leftarrow \text{argsort}(V, \text{desc}), \quad \hat{I}_V \leftarrow \text{argsort}(V, \text{asc})$;

11    $\pi \leftarrow [\,]$;

12    **for** $i = 1$ **to** $N$ **do**

13      append $\check{I}_V[i]$ to $\pi$;

14      append $\hat{I}_V[(i-1)(n-1) + 1 : i(n-1)]$ to $\pi$;

15    **return** $\pi$;

---

arrangement. Concretely, we first sort the columns by their mean absolute value and partition them along the column dimension into small subgroups. Then, the subgroup with the largest average variance is paired with subgroups of the smallest variance to form the first group, and this process is repeated until all subgroups are assigned.

Second, after obtaining the permutation order $\pi$, we construct the corresponding permutation matrix $P$, defined as follows:

$$P_{ij} = \begin{cases} 1, & \text{if } i = \pi(j), \\ 0, & \text{otherwise.} \end{cases}, \text{ where } P \in \{0, 1\}^{n \times n} \tag{10}$$

Lastly, we fuse the permutation matrix into the weight parameters to eliminate additional online computation. For the Phi-mini-MoE-Instruct and Qwen3-30B-A3B models, the permutation is applied in both the self-attention and MoE-FFN blocks. In the self-attention block, we multiply the permutation matrix with $W_{R;V}$ and apply its transpose to $W_{R;out}$. In the MoE-FFN block, the permutation matrix is multiplied with $W_{R;up}$, while its transpose is applied to $W_{R;down}$ for each expert.

For DeepSeek-V2-Lite, the permutation is applied to all experts, including the shared expert, in the MoE-FFN block. Specifically, $W_{R;up}$ is multiplied by the permutation matrix, and $W_{R;down}$ is multiplied by its transpose for every expert. In the self-attention block, due to the unique structure of the DeepSeek family, additional steps are required to preserve output invariance. First, the layer normalization is absorbed into the weight matrix. Then, we decompose

$$W_{R;kv\_a} = \begin{bmatrix} W_{R;compressed\_kv}, & W_{R;k\_pe} \end{bmatrix}$$

into $W_{R;compressed\_kv}$ and $W_{R;k\_pe}$. The permutation matrix is multiplied with $W_{R;compressed\_kv}$, while the transpose of the permutation matrix is applied to $W_{R;kv\_b}$ to preserve output invariance.

### A.2.2 ANALYSIS

POG is designed to reduce weight clustering error under the Block-wise setting, where weights are partitioned into clustering groups (of fixed size $g$) along the embedding dimension clustered independently. In this regime, a column permutation changes which weight values are placed into the same group, and therefore can change the within-group distribution and the resulting K-means solutions. By constructing a permutation matrix that reorders columns with different variability, POG aims to form groups that are better conditioned for clustering.

Table 7: Performance in perplexity (PPL) on Wiki2 and C4 dataset, and accuracy on Arc-Challenge (A-c), Arc-easy (A-e), HellaSwag (HS), MMLU (ML), PIQA (PQ) and WinoGrande (WG). Code-Quant is set as A8W4 Embedding-wise. We report the BF16 baseline in the first row, and mark the methods as BF16.

| | Models | Methods | Wiki2 (↓) | C4 (↓) | A-c (↑) | A-e (↑) | HS (↑) | ML (↑) | PQ (↑) | WG (↑) | Avg (↑) |
|---|---|---|---|---|---|---|---|---|---|---|---|
| **A8W4 Embedding-wise** | Phi-mini-MoE-Instruct | BF16 | 6.83 | 13.06 | 0.581 | 0.813 | 0.759 | 0.681 | 0.797 | 0.753 | 0.731 |
| | | RTN | 12.13 | 20.46 | 0.460 | 0.730 | 0.618 | 0.497 | 0.741 | 0.632 | 0.613 |
| | | SqueezeLLM | 7.41 | **13.65** | 0.565 | 0.795 | 0.736 | 0.658 | 0.791 | **0.746** | 0.715 |
| | | SmoothQuant | 9.50 | 16.23 | 0.481 | 0.741 | 0.653 | 0.569 | 0.756 | 0.638 | 0.634 |
| | | QuaRot | 7.69 | 14.15 | 0.549 | 0.787 | 0.735 | 0.652 | 0.786 | 0.737 | 0.708 |
| | | **CodeQuant** | **7.36** | 13.73 | **0.579** | **0.796** | **0.741** | **0.668** | **0.796** | 0.732 | **0.719** |
| | Qwen3-30B-A3B | BF16 | 9.04 | 14.05 | 0.566 | 0.793 | 0.776 | 0.778 | 0.805 | 0.694 | 0.735 |
| | | RTN | 14.09 | 21.65 | 0.284 | 0.446 | 0.692 | 0.643 | 0.656 | 0.626 | 0.558 |
| | | SqueezeLLM | **9.37** | **14.56** | 0.529 | 0.768 | 0.743 | **0.764** | 0.770 | 0.671 | 0.707 |
| | | SmoothQuant | 11.77 | 17.82 | 0.463 | 0.703 | 0.721 | 0.695 | 0.773 | 0.667 | 0.670 |
| | | QuaRot | 11.18 | 16.58 | 0.471 | 0.671 | 0.696 | 0.708 | 0.766 | 0.654 | 0.661 |
| | | **CodeQuant** | 9.81 | 15.11 | **0.535** | **0.779** | **0.754** | 0.757 | **0.797** | **0.679** | **0.717** |
| | DeepSeek-V2-Lite | BF16 | 6.69 | 9.32 | 0.491 | 0.759 | 0.780 | 0.551 | 0.804 | 0.709 | 0.682 |
| | | RTN | 7.72 | 10.89 | 0.469 | 0.719 | 0.732 | 0.457 | 0.790 | 0.671 | 0.640 |
| | | SqueezeLLM | 6.93 | 9.60 | 0.485 | 0.755 | 0.760 | 0.525 | **0.803** | 0.701 | 0.658 |
| | | SmoothQuant | 7.61 | 10.70 | 0.457 | 0.729 | 0.754 | 0.480 | 0.794 | 0.674 | 0.648 |
| | | QuaRot | 7.29 | 10.08 | 0.466 | 0.737 | 0.757 | 0.493 | 0.792 | 0.705 | 0.658 |
| | | **CodeQuant** | **6.84** | **9.50** | **0.487** | **0.764** | **0.773** | **0.533** | 0.798 | **0.709** | **0.678** |
| **A8W4 Block-wise** | Phi-mini-MoE-Instruct | RTN | 8.68 | 14.93 | 0.530 | 0.777 | 0.683 | 0.578 | 0.770 | 0.671 | 0.668 |
| | | SqueezeLLM | 7.18 | 13.46 | **0.576** | 0.801 | **0.744** | **0.670** | **0.797** | **0.759** | **0.724** |
| | | SmoothQuant | 8.40 | 14.67 | 0.516 | 0.768 | 0.697 | 0.602 | 0.769 | 0.688 | 0.673 |
| | | QuaRot | 7.48 | 13.65 | 0.550 | 0.794 | 0.737 | 0.645 | 0.786 | 0.737 | 0.708 |
| | | **CodeQuant** | **7.11** | **13.33** | 0.575 | **0.817** | **0.744** | 0.661 | 0.792 | 0.751 | 0.723 |
| | DeepSeek-V2-Lite | RTN | 7.47 | 10.40 | 0.455 | 0.743 | 0.764 | 0.488 | 0.788 | 0.687 | 0.654 |
| | | SqueezeLLM | 6.86 | 9.53 | 0.469 | 0.754 | 0.773 | **0.535** | 0.796 | 0.706 | 0.672 |
| | | SmoothQuant | 7.42 | 10.39 | 0.459 | 0.724 | 0.748 | 0.476 | 0.792 | 0.665 | 0.644 |
| | | QuaRot | 7.22 | 10.03 | 0.466 | 0.746 | 0.760 | 0.508 | 0.795 | 0.688 | 0.661 |
| | | **CodeQuant** | **6.83** | **9.49** | **0.472** | **0.756** | **0.775** | **0.535** | **0.804** | **0.708** | **0.675** |

In contrast, under the Embedding-wise setting where clustering is performed over the entire embedding dimension as a single set (i.e., without block partitioning), POG has no effect on the clustering result. This is because permutation only reorders the elements of the set being clustered, and K-means is order-invariant. In Block-wise setting, however, each block is formed by selecting a subset of elements along with the embedding dimension after permutation. As a result, the composition of each subset depends on the ordering, and thus permutation can change which elements belong to the same block. Consequently, POG will only benefit Block-wise setting.

## A.3 HARDWARE EVALUATION SETTINGS

We use Accel-Sim (Khairy et al., 2020), a state-of-the-art open-source GPU simulator, and modify its configuration and trace files to model both the original RTX A100 80GB GPU and an A100 with CodeQuant-optimized tensor cores, as shown in Section 3.4. The simulator is calibrated against real A100 measurements, achieving less than 1% latency error, consistent with prior GPU module design studies (Mo et al., 2025; Guo et al., 2023; Avalos Baddouh et al., 2021). We configure tensor cores with a matrix multiplication size of $16 \times 4 \times 8$ and 64 shared memory banks to improve lookup table reuse and reduce bank conflicts.

## A.4 CODEQUANT A8W4 EMBEDDING-WISE ACCURACY PERFORMANCE

With 8-bit activation quantization, the error of activation quantization is minimized. In this testing setup, the effectiveness of quantization/clustering method will stand out. Table 7 summarizes the main results of offline version CodeQuant under Embedding-wise setting. Our method is generally better than all baselines. Meanwhile, we notice that SqueezeLLM also performs well in this case showing the promising of clustering method. Table 8 summarizes the comparison between CodeQuant$_{had}$, SpinQuant and DuQuant. As we discussed in the main content, with more layers being quantized or clustered in this online scenario, the accuracy is lower than Table 7, but we make the same quantization/clustering layers setup to make sure fair comparison. Compared with these two strong rotation methods, our algorithm excels them proving the effectiveness of CodeQuant.

Table 8: Rotation-based method performance comparison. CodeQuant$_{had}$ indicates that online Hadamard transforms are enabled during the quantization process.

| | Models | Methods | Wiki2 ($\downarrow$) | C4 ($\downarrow$) | A-c ($\uparrow$) | A-e ($\uparrow$) | HS ($\uparrow$) | ML ($\uparrow$) | PQ ($\uparrow$) | WG ($\uparrow$) | Avg ($\uparrow$) |
|---|---|---|---|---|---|---|---|---|---|---|---|
| A8W4 Embedding-wise | DeepSeek-V2-Lite | DuQuant | 8.43 | 11.94 | 0.455 | 0.723 | 0.728 | 0.406 | 0.815 | 0.665 | 0.632 |
| | | SpinQuant$_{had}$ | 8.62 | 11.95 | 0.431 | 0.696 | 0.726 | 0.432 | 0.781 | 0.650 | 0.619 |
| | | **CodeQuant$_{had}$** | **7.80** | **10.85** | **0.457** | **0.726** | **0.742** | **0.472** | 0.787 | **0.676** | **0.643** |
| | Qwen3-30B-A3B | DuQuant | 12.70 | 18.69 | 0.455 | **0.752** | 0.604 | 0.685 | **0.777** | 0.622 | 0.649 |
| | | SpinQuant$_{had}$ | 13.55 | 19.53 | 0.474 | 0.716 | 0.669 | 0.677 | 0.762 | 0.616 | 0.652 |
| | | **CodeQuant$_{had}$** | **10.98** | **16.80** | **0.481** | 0.725 | **0.719** | **0.702** | 0.766 | **0.650** | **0.674** |

## A.5 LUT KERNEL PERFORMANCE ON CPU

Table 9: Latency and Memory Evaluation on CPU

| Bit Width | Method | Phi-mini-MoE-Instruct | | DeepSeek-V2-Lite | | Qwen3-30B-A3B | |
|---|---|---|---|---|---|---|---|
| | | Mem. (GB) $\downarrow$ | Lat. (s) $\downarrow$ | Mem. (GB) $\downarrow$ | Lat. (s) $\downarrow$ | Mem. (GB) $\downarrow$ | Lat. (s) $\downarrow$ |
| BF16 | Llama.cpp (CPU) | 14.3 | 40.1 | 29.3 | 50.0 | 56.9 | 66.1 |
| A8W4 | Llama.cpp (CPU) | **4.1** | 15.0 | **8.8** | 17.1 | **16.2** | 20.1 |
| | **CodeQuant (CPU)** | **4.1** | **13.3** | 8.9 | **14.2** | 16.5 | **15.9** |

T-MAC (Wei et al., 2025) implements mixed-precision GEMM via a lookup table–based kernel within the Llama.cpp framework (Gerganov & ggml-org contributors, 2023), enabling efficient CPU execution. We evaluate CodeQuant by benchmarking the A8W4 T-MAC kernel against BF16 and A8W4 models in Llama.cpp. The experiments are conducted on an Intel(R) Xeon(R) w7-3445 CPU (Intel Corporation, 2025) using 20 threads. On CPU, CodeQuant achieves up to $4.15\times$ speedup over BF16 baselines and consistently outperforms the quantization baselines. The gains are larger on CPU than on GPU primarily because CPU inference exposes less parallelism and is more memory-bound (Wei et al., 2025), making the improvements over the baseline more pronounced. In addition, efficient LUT instructions on CPUs further amplify CodeQuant's advantage over quantized baselines.

Table 10: Impact of POG

| Method | Phi-mini-MoE-Instruct | | | DeepSeek-V2-Lite | | |
|---|---|---|---|---|---|---|
| | Wiki2 $\downarrow$ | C4 $\downarrow$ | Acc $\uparrow$ | Wiki2 $\downarrow$ | C4 $\downarrow$ | Acc $\uparrow$ |
| W/O POG | 7.31 | 13.66 | 0.710 | 7.08 | 9.88 | 0.663 |
| **W/ POG** | **7.28** | **13.54** | **0.714** | **7.03** | **9.79** | **0.668** |

Table 11: KL Penalty Impact on Router

| Model | Method | Change Rate (%) $\downarrow$ |
|---|---|---|
| DeepSeek-V2-Lite | QuaRot | 41.47 |
| | CodeQuant w/o KL | 24.33 |
| | **CodeQuant w/ KL** | **22.82** |
| Qwen3-30B-A3B | QuaRot | 72.15 |
| | CodeQuant w/o KL | 60.21 |
| | **CodeQuant w/ KL** | **59.58** |

## A.6 IMPACT OF POG

We evaluate the impact of POG operation on Phi-mini-MoE-Instruct and DeepSeek-V2-Lite under the A4W4 Block-wise configuration with a fixed group size of $g = 1024$. As shown in Table 10, removing permutation consistently degrades performance. On Phi-mini-MoE-Instruct, with POG applied, perplexity increases by 0.03 on WikiText2 and 0.12 on C4, while accuracy drops by 0.4%. A similar pattern is observed on DeepSeek-V2-Lite, confirming the generality of this effect.

## A.7 IMPACT OF KL PENALTY ON ROUTER LOGITS

We measure the effect of KL divergence on router stability for DeepSeek-V2-Lite and Qwen3-30B-A3B under the A4W4 Embedding-Wise setting. The change rate is defined as the layer-wise average change in Top-$K$ expert indices (with $K = 6$ for DeepSeek-V2-Lite and $K = 8$ for Qwen3-30B-A3B) computed by comparing the router outputs before and after quantization. Results are averaged over 50 samples from the WikiText-2 test set.

As shown in Table 11, adding the KL penalty consistently reduces routing perturbation. On DeepSeek-V2-Lite, the change rate drops from 24.33% to 22.82%. A similar trend is observed on Qwen3-30B-A3B, where KL regularization yields a reduction from 60.21% to 59.58%, despite its larger 128-expert

MoE blocks. These results indicate that KL regularization helps preserve the expert-routing pattern during quantization and mitigates performance degradation.

