# OpenReview forum: "CodeQuant: Unified Clustering and Quantization for Enhanced Outlier Smoothing in Low-Precision Mixture-of-Experts"
_ICLR.cc/2026/Conference — ICLR 2026 Poster_

### Official Review · Reviewer_BbsM · 2025-10-25

**Soundness:** 2
**Presentation:** 3
**Contribution:** 3
**Rating:** 6
**Confidence:** 4

**Summary:**

This paper investigates the efficiency of MoE-based LLMs and proposes a vector quantization method named CodeQuant, which incorporates learnable rotation and clustering operations to achieve competitive W4A4 performance. The authors further design optimized LUT kernels to demonstrate improvements in execution latency.

**Strengths:**

1. The proposed hardware-oriented implementation effectively enhances the efficiency of MoE-based LLMs.
2. The specially designed component for MoE, such as the KL loss term, is validated to be effective through ablation studies.

**Weaknesses:**

1. Since learnable rotation has been extensively explored in prior LLM post-training quantization (PTQ) works such as SpinQuant and OSTQuant, the paper should more clearly articulate its key differences and advantages over these baselines.
2. Some of the compared baselines appear outdated and do not represent the current state-of-the-art.
3. The proposed kernel is evaluated only within the Accel-Sim framework, lacking validation on real hardware

**Questions:**

1. What is the additional training cost introduced by the learnable rotation matrix?
2. Could you include comparisons with more competitive rotation-based quantization baselines mentioned in the Experiments section, such as DuQuant and SpinQuant?
3. Could you report actual latency and memory measurements on real Nvidia GPUs (e.g., RTX 4090 or A100) to support the Accel-Sim results?

---

> ### Author Response · Authors · 2025-11-21
>
> We thank the reviewer for the insightful comments. Below, we summarize each question and address them individually:
>
> **Questions1: What is the additional training cost introduced by the learnable rotation matrix?**
>
> **A**: The additional cost of learning the rotation matrix is minimal. In our implementation, the AOS stage is optimized for a fixed number of calibration steps (128 steps), during which only the rotation parameters are updated. Specifically, on an H100 GPU the AOS stage takes approximately **15/20/30 minutes** for Phi‑mini‑MoE‑Instruct, DeepSeek‑V2‑Lite, and Qwen3‑30B‑A3B, respectively. The subsequent ACCF stage takes about **30/40/110 minutes** for the same three models respectively. At inference time, all operations are fused into the model weights, resulting in zero additional overhead.
>
> **Questions2: Could you include comparisons with more competitive rotation-based quantization baselines mentioned in the Experiments section, such as DuQuant and SpinQuant?**
>
> **A**: Table2 in the revised paper contains the DuQuant and SpinQuant experiments on Qwen3-30B-A3B and Deepseek-V2-Lite. Both experiments are performed using A4W4 Embedding-wise setup. The notation for downstream tasks is the same as our paper (i.e., Arc-Challenge(A-c), Arc-easy (A-e), HellaSwag (HS), MMLU (ML), PIQA (PQ) and WinoGrande (WG)). To ensure fair comparison, we add online Hadamard transformations for CodeQuant, refer to $\text{CodeQuant}_{had}$ in the revision paper. The experiment results on Qwen3-30B-A3B is reported in the following table, our method achieved higher average accuracy than both SpinQuant (0.653 vs. 0.590) and DuQuant (0.653 vs. 0.637). Overall, the results indicate that our method provides a more effective quantization strategy than prior rotation-only approaches.
>
> Table1: Evaluation of PTQ Quantization Methods on Qwen3-30B-A3B
> | Method        | Wiki2      | C4    | A-c   | A-e   | HS    | ML    | PQ    | WG    | Avg   |
> | ------------- | ---------- | ----- | ----- | ----- | ----- | ----- | ----- | ----- | ----- |
> | DuQuant       | 13.52      | 20.10 | 0.472 | 0.662 | 0.687 | 0.654 | 0.739 | 0.606 | 0.637 |
> | $\text{SpinQuant}_{had}$ | 14.61      | 22.07 | 0.415 | 0.600 | 0.628 | 0.584 | 0.692 | 0.622 | 0.590 |
> | $\text{CodeQuant}_{had}$| 12.69      | 19.89 | 0.477 | 0.697 | 0.691 | 0.679 | 0.739 | 0.635 | 0.653 |
>
> **Questions3: Could you report actual latency and memory measurements on real Nvidia GPUs (e.g., RTX 4090 or A100) to support the Accel-Sim results?**
>
> **A**: In the paper, we propose the LUT based GEMM kernel that targets the future GPU architecture with dedicated hardware support. To prove the feasibility of the proposed design in current hardware, we evaluate a LUT-based kernel on different models in an A100 GPU with the A4W4 Embedding-wise settings detailed on Section 4. We report the measured latency and memory of CodeQuant in the table below.
>
> Table2: Comparison of Latency and Memory on A100.
> | | Phi-mini-MoE-Instruct       |    | DeepSeek-V2-Lite        |     | Qwen3-30B-A3B            |            |  Average Improvement       |               |
> | ------------------------------- | ----------- | ---------- | ----------- | ---------- | ----------- | ---------- | ------- | ------------- |
> |                                 | Latency(ms) | Memory(GB) | Latency(ms) | Memory(GB) | Latency(ms) | Memory(GB) | Speedup | Memory Saving |
> | BF16                            | 9756        | 14.3       | 15548       | 29.3       | 22176       | 56.9       | 1x      | 1x            |
> | QuaRot                          | 4646        | 4.1        | 6478        | 8.8        | 8529        | 16.2       | 2.36x   | 3.44x         |
> | SquezeLLM                       | 4878        | 4.1        | 7404        | 8.9        | 9642        | 16.3       | 2.13x   | 3.42x         |
> | CodeQuant                      | 4358       | 4.1       | 6212        | 8.9        | 8215        | 16.3       | 2.48x   | 3.42x         |
>
> As shown in the table, the CodeQuant kernel achieves an average of 2.48x speedup across different models due to less computation and memory overhead of low-bit weights and activations. In the Accel-Sim GPU results reported in Section 4.2, Codequant achieves an average of 2.63x speedup, which is higher than the real GPU evaluation results. This is because in our proposed design in Section 3.4, we target the future GPU architecture with memory layout that is more suitable for LUT operations and support from dedicated hardware units. And the Accel-Sim settings reflect such hardware change that does not exist in real A100 GPU, leading to a larger speedup compared to real GPU evaluation results. With more advanced hardware support to accelerate the table lookup operation, we expect Codequant to achieve larger speedup.

---

> ### Author Response · Authors · 2025-11-21
>
> **Weakness1: Since learnable rotation has been extensively explored in prior LLM post-training quantization (PTQ) works such as SpinQuant and OSTQuant, the paper should more clearly articulate its key differences and advantages over these baselines.**
>
> **A**: Prior LLM PTQ methods such as SpinQuant and OSTQuant have already explored learnable rotation matrices. SpinQuant proposes to parameterize the rotation matrix via the Cayley transform and fine-tune this rotation end-to-end during post-training quantization. OSTQuant further introduces a new KL-based objective, the so-called KL-top loss, to end-to-end optimize both the rotation matrices and the smoothing scales.
>
> Our approach differs from these methods in two key ways. First, instead of relying on a global KL divergence at the output level like OSTQuant, we learn rotations using a sum of per-layer MSE losses between full-precision and quantized hidden states. This directly improves layer-wise reconstruction quality and avoids the instability that often arises from matching logits globally. Second, we do not fine-tune the full model together with the rotations which is adopted by QpinQuant. The rotation parameters are learned in a lightweight calibration stage with the rest of the network frozen, which makes our procedure substantially more computationally efficient than the end-to-end optimization used in SpinQuant and OSTQuant, while still delivering competitive or superior quantization performance.
>
> We already provide a performance comparison with SpinQuant in our revision of paper (Table 2), and we also provide an OSTQuant comparison here. Due to resource limitations, we fine-tuned only the last three layers of Qwen3-30B-A3B using OSTQuant and Codequant, respectively. Even under this restricted setting, Codequant still outperforms OSTQuant.
>
> |Method|Wiki2|C4|A-c|A-e|HS|ML|PQ|WG|Avg|
> |---|---|---|---|---|---|---|---|---|---|
> |OSTQuant|13.68|20.00|0.466|0.700|0.696|0.694|0.750|0.660|0.661|
> |CodeQuant|9.19|14.32|0.519|0.796|0.662|0.775|0.821|0.697|0.712|
>
> **Weakness2: Some of the compared baselines appear outdated and do not represent the current state-of-the-art.**
>
> **A**: Thank you for raising this point, this feedback helps strengthen the impact and fairness of our evaluation. In response, we have added comparisons with two influential rotation-based PTQ methods, DuQuant and SpinQuant, and included the results in the revised paper (Table 2). We also provide a detailed analysis using Qwen3-30B-A3B as an example in Question 2 of the rebuttal, and we refer the reviewer there for a more complete discussion. These additions ensure that CodeQuant is evaluated against strong, modern baselines and demonstrate that our method remains competitive under the same quantization configuration.
>
> **Weakness3: The proposed kernel is evaluated only within the Accel-Sim framework, lacking validation on real hardware.**
>
> **A**: Thank you for raising this point, as it helps strengthen the practical relevance of our work. We provide detailed real-hardware measurements and analysis in Question 3 of the rebuttal, which demonstrate the feasibility of our kernel beyond simulation.

---

> > ### Author Response · Authors · 2025-11-27
> >
> > Dear reviewer BbsM, please let us know if our response has addressed your questions, and feel free to let us know if you have any additional questions. Thank you.

---

### Official Review · Reviewer_8Fns · 2025-10-29

**Soundness:** 2
**Presentation:** 2
**Contribution:** 2
**Rating:** 6
**Confidence:** 2

**Summary:**

The paper introduces CodeQuant, a quantization method for MoE models. It combines Activation-oriented Outlier Smoothing (AOS), Adaptive Weight Clustering with Centroid Finetuning (ACCF), and Permutation-Invariant Outlier Grouping (POG), and provides kernels for LUT-driven execution. The experiments evaluate both quantization accuracy and latency.

**Strengths:**

1. AOS directly optimizes activation quantization error on calibration data. The use of Cayley transforms to enforce orthogonality is simple and numerically stable compared with naive orthogonalization schemes.

2. ACCF alternates optimization over centroids and assignments, which is well motivated. The additional KL term for router logits is also reasonable and conceptually sound.

3. POG targets a very practical issue: improving block-wise clustering quality under tight centroid budgets.

4. The LUT-based kernel is evaluated via simulation to estimate speedups and memory savings, and the reported numbers appear reasonable.

5. Overall, the motivation is clear, and both the method and the experimental design are appropriate to support the paper’s claims.

**Weaknesses:**

1. Some of the bolded entries in the tables appear incorrect. For example: (i) W4A4 Embedding, Phi-mini-MoE-Instruct, A-c dataset; (ii) W8A4 Embedding, Phi-mini-MoE-Instruct, WG dataset.

2. The GPU evaluation of the proposed method is based solely on simulation, whereas QuaRot and SqueezeLLM are benchmarked on real GPUs. In addition, there may be non-trivial overhead, especially from POG, which could introduce extra computation or memory traffic. The authors should discuss these potential costs in more detail.

3. The experiments are limited to perplexity and commonsense QA benchmarks. Additional benchmarks such as GSM8K or MATH would provide a more comprehensive assessment of the method’s impact on reasoning-heavy tasks.

4. The paper does not report the pre-processing (quantization runtime). Including this would help clarify the practical cost of applying the method in real deployments.

**Questions:**

See weaknesses above.

---

> ### Author Response · Authors · 2025-11-21
>
> We thank the reviewer for the insightful comments. Below, we summarize each question and address them individually:
>
> **Weakness1: Some of the bolded entries in the tables appear incorrect. For example: (i) W4A4 Embedding, Phi-mini-MoE-Instruct, A-c dataset; (ii) W8A4 Embedding, Phi-mini-MoE-Instruct, WG dataset.**
>
> **A**: We thank the reviewer for pointing out the formatting issues. Some bolded entries in the tables were incorrectly formatted, although the underlying numerical values are correct. We have fixed all formatting errors in the revised manuscript.
>
> **Weakness2: The GPU evaluation of the proposed method is based solely on simulation, whereas QuaRot and SqueezeLLM are benchmarked on real GPUs. In addition, there may be non-trivial overhead, especially from POG, which could introduce extra computation or memory traffic. The authors should discuss these potential costs in more detail.**
>
> **A**: The POG step described in Figure 3(c) is carried out offline, where the permutation order is determined in advance. During runtime, the weight matrices remain fixed, and inference proceeds in the same way as a conventional MoE. As a result, this approach does not introduce any additional computation or memory traffic during inference. Additionally, after the centroids are obtained by minimizing ACCF, each weight is assigned to its nearest centroid using the distance metric defined in Equation 7 of the paper, and there is no computation overhead for CodeQuant during inference.
>
> In the revised version, we include real GPU measurements on an A100 to complement the simulation result. We benchmark the CodeQuant LUT-based kernel on various models using the A4W4 embedding-wise settings described in Section 4. The table below reports the measured latency and memory usage of CodeQuant together with those of the baseline methods. We also report the average speedup and memory savings across different models with the numbers normalized to the latency and memory usage of FP16 baseline.
>
> Table1: Comparison of Latency and Memory across Models on A100.
> | | Phi-mini-MoE-Instruct       |    | DeepSeek-V2-Lite        |     | Qwen3-30B-A3B            |            |  Average Improvement       |               |
> | ------------------------------- | ----------- | ---------- | ----------- | ---------- | ----------- | ---------- | ------- | ------------- |
> |                                 | Latency(ms) | Memory(GB) | Latency(ms) | Memory(GB) | Latency(ms) | Memory(GB) | Speedup | Memory Saving |
> | BF16                            | 9756        | 14.3       | 15548       | 29.3       | 22176       | 56.9       | 1x      | 1x            |
> | QuaRot                          | 4646        | 4.1        | 6478        | 8.8        | 8529        | 16.2       | 2.36x   | 3.44x         |
> | SquezeLLM                       | 4878        | 4.1        | 7404        | 8.9        | 9642        | 16.3       | 2.13x   | 3.42x         |
> | CodeQuant                      | 4358       | 4.1       | 6212        | 8.9        | 8215        | 16.3       | 2.48x   | 3.42x         |
>
> As shown in the table, the CodeQuant kernel achieves an average of 2.48x speedup compared to the FP16 baseline due to less computation and memory overhead from low-bit weights and activations. Compared to baseline methods, CodeQuant can still achieve a better performance in terms of speedup. With more advanced hardware support to accelerate the table lookup operation, we expect CodeQuant to achieve larger speedup.

---

> ### Author Response · Authors · 2025-11-21
>
> **Weakness3: The experiments are limited to perplexity and commonsense QA benchmarks. Additional benchmarks such as GSM8K or MATH would provide a more comprehensive assessment of the method’s impact on reasoning-heavy tasks.**
>
> **A**: We evaluate DeepSeek-V2-Lite and Qwen3-30B-A3B under the A4W4 setup on both GSM8K and MATH500, and include the new evaluation in the revision of paper. For GSM8K, we use an 8-shot setting, and for MATH500, we use a 4-shot setting, where k-shot refers to providing the model with k worked examples in the prompt before answering each test question. We observe that CodeQuant consistently outperforms QuaRot under the same bit width.
>
> The accuracies of MATH500 on DeepSeek-V2-Lite may seem low, but they align with the BF16 results reported in the DeepSeek-V2 technical report (DeepSeek-AI et al., 2024). The slightly lower numbers in our reproduction stem from minor variations in prompt formatting, a known factor that can noticeably influence few-shot math-reasoning performance.
>
> Table1: Math Reasoning Performance on DeepSeek-V2-Lite and Qwen3-30B-A3B
> | Model            | Method    | GSM8K (8-shot) | MATH500 (4-shot) |
> | ---------------- | --------- | --------------- | ----------------- |
> | DeepSeek-V2-Lite | BF16      | 0.364           | 0.121             |
> | | QuaRot           | 0.231     | 0.093           |
> | | **CodeQuant**    | **0.330** | **0.108**       |
> | Qwen3-30B-A3B    | BF16      | 0.924           | 0.322             |
> | | QuaRot           | 0.508     | 0.128           |
> | | **CodeQuant**    | **0.867** | **0.241**       |
>
> **Weakness4: The paper does not report the pre-processing (quantization runtime). Including this would help clarify the practical cost of applying the method in real deployments.**
>
> **A**: In the revision of the paper, we include the pre-processing runtime in lines 361–365. Specifically, the AOS stage requires approximately 15/20/30 minutes for Phi-mini-MoE-Instruct, DeepSeek-V2-Lite, and Qwen3-30B-A3B on H100 GPUs, respectively. The subsequent ACCF stage requires 30/40/110 minutes for the same models. At inference time, all operations (rotation, clustering, and permutation) introduce no additional overhead.

---

> > ### Author Response · Authors · 2025-11-27
> >
> > Dear reviewer 8Fns, please let us know if our response has addressed your questions, and feel free to let us know if you have any additional questions. Thank you.

---

### Official Review · Reviewer_yk1W · 2025-10-30

**Soundness:** 3
**Presentation:** 2
**Contribution:** 3
**Rating:** 4
**Confidence:** 4

**Summary:**

This paper proposes a codebook-based clustering method for quantizing Mixture-of-Experts (MoE) models. To mitigate the outlier issue, it combines 1) activation smoothing, which transfers outliers to weights, and 2) a training-based clustering method to reduce weight quantization error. The paper focuses on MoE LLMs, and the training loss is carefully designed to align the token routing behavior of the quantized model with the original model.  The authors validated inference performance benefit via a simulation framework and proposed insights for hardware and software co-design.

**Strengths:**

-  The work focus the timely and challenging problem of quantizing Mixture-of-Experts (MoE) models. The training loss is well-tailored for these architectures, giving it practical significant relevance for industry deployment.
- It provides a analysis on the computational patterns for clustering-based GEMM operations, offering a perspective for future hardware and software co-design.

**Weaknesses:**

- The technical innovation appears limited, as the method primarily combines previously established techniques of outlier transformation and clustering.
- The benchmarking and accuracy evaluation are conducted on relatively small models. Since MoE models are typically very large, validating the method on larger-scale models (e.g., Qwen-235B, DeepSeek-R1) would strengthen the claims. Furthermore, evaluation is limited to next-token prediction tasks; testing on more complex reasoning benchmarks (e.g., GSM8K, MATH500) would be more compelling for modern LLM capabilities.
- While inference performance is compared, a more detailed breakdown is lacking. It would be beneficial to provide detailed testing parameters and analyze performance separately for the prefill and decoding stages, as they have different computational characteristics (memory bandwidth vs. compute density).

**Questions:**

- What is the quantitative impact of the KL loss on final accuracy? Is there evidence that preserving the original token-expert assignment is necessary after quantization alters the weights?
- Computational Overhead:** What percentage of the total inference time is spent on the clustering operations? How does the performance benefit differ between the prefill and decoding stages?

---

> ### Author Response · Authors · 2025-11-21
>
> We thank the reviewer for the insightful comments. Below, we summarize each question and address them individually:
>
> **Questions1: What is the quantitative impact of the KL loss on final accuracy? Is there evidence that preserving the original token-expert assignment is necessary after quantization alters the weights?**
>
> **A**: Table 3 in the paper (Table 5 in the revised version) provides a direct ablation showing that adding the KL term improves performance under the A4W4 Block-wise configuration, including lower WikiText2 and C4 perplexity as well as higher average accuracy. This demonstrates that preserving the original token–expert distribution after applying CodeQuant leads to measurable performance improvements. This is because the KL regularization reduces router index drift, keeping the post-quantization routing pattern closer to the original model, which in turn stabilizes expert utilization and improves downstream accuracy. This observation is aligned with previous works that discuss quantization on MoE (e.g. MoEQuant). In the revised version of the paper, we further confirm this observation on DeepSeek-V2-Lite (Table 5 in the revision paper), where we see the same trend.

---

> ### Author Response · Authors · 2025-11-21
>
> **Questions2: Computational Overhead: What percentage of the total inference time is spent on the clustering operations? How does the performance benefit differ between the prefill and decoding stages?**
>
> **A**: The clustering step is performed offline. After the centroids are obtained by minimizing ACCF, each weight is assigned to its nearest centroid using the distance metric defined in Equation 7 of the paper, and there is **no computation overhead** for the clustering during inference.
>
> Moreover, we profile the latencies of the LUT-based kernel executing an activation and weight matrix multiplication on an A100 GPU. The activation matrix size is [4096, 4096] and the weight matrix size is [4096, 4096], representing one linear layer in the model with a hidden dimension of 4096 processing a sequence with 4096 tokens. The activation and weight matrices are quantized to 4-bit. The latency for this matrix multiplication is 21ms. The computation of centroids multiplying activation, as shown in Step 1 of Figure 4 (b), takes 34% of the total latency. The output lookup shown in Step 2 of Figure 4 (b), takes 66% of the total latency. Compared to FP16 matrix multiplication, the 4-bit CodeQuant matrix multiplication achieves a 2.4x speedup.
>
> We evaluate the latency performance of W4A4 Embedding-wise CodeQuant under different prefilling and decoding settings on the Phi-mini-MoE-Instruct model. First, we set the prefilling token length as 512 and the decoding token length as 128, 1024, 2048, and 4096. The batch size is set as 16. The table below shows the results.
>
> Table1: Latency and Speedup Across Decoding Lengths
> | Decoding Length|128|||1024|||2048|||4096 |||
> |-|-|-|-|-|-|-|-|-|-|-|-|-|
> |**Stage**|**prefill**|**decode**|**total**|**prefill**|**decode**|**total**|**prefill**|**decode**|**total**|**prefill**|**decode**|**total**|
> | BF16 latency(ms)|952|8804|9756|968|72164|73132|985|161314|162299|971|391047|392018|
> | CodeQuant latency(ms)|426|3932|4358|432|31931|32363|442|69833|70275|430|166403|166833|
> | CodeQuant speedup|2.23|2.24|2.24|2.24|2.26|2.26|2.23|2.31|2.31|2.26|2.35|2.35|
>
> The decoding stage is memory bound because there will be only one current token being computed with the model weights and KV cache is enabled to represent information from previous tokens. In this case, the GPU is executing matrix-vector multiplication that under utilizes the computation power GPU can provide and the main overhead comes from the model weight and KV cache data movement. As the decoding sequence length increases, the KV cache size as well as the number of model weight movements increase, making the memory movement overhead more significant. So, in the decoding stage, the memory reduction from model weight clustering and activation quantization in CodeQuant brings larger speedup when the decoding length increases, as shown in the table above.
>
> In the second experiment, we set the decoding length as 128, and the prefilling token length as 512, 1024, 2048, and 4096. The table below shows the results. As the prefilling sequence length increases, the speedup of CodeQuant also increases, as it reduces the computation and memory movement overhead. The speedup in the prefilling stage is a little lower than that of the decoding stage, mainly because all the tokens are processed in parallel in the prefilling stage, which means the weights can be reused by more activations with one data movement. So the overhead of memory movement is less significant in the prefilling stage compared to the decoding stage. CodeQuant achieves a significant speedup in the prefilling stage, although the benefits of data movement saving from the weight clustering is marginally less compared to the decoding stage.
>
> Table2: Latency and Speedup Across Prefilling Lengths
> |Prefilling Length|512|||1024|||2048|||4096|||
> |-|-|-|-|-|-|-|-|-|-|-|-|-|
> |**Stage**|**prefill**|**decode**|**total**|**prefill**|**decode**|**total**|**prefill**|**decode**|**total**|**prefill**|**decode**|**total**|
> |BF16 latency(ms)|952|8804|9756|1327|9554|10880|2138|11691|13829|3612|16645|20258|
> |CodeQuant latency(ms)|426|3932|4358|589|4227|4817|930|5061|5990|1557|7114|8671|
> |CodeQuant speedup|2.23|2.24|2.24|2.25|2.26|2.26|2.30|2.31|2.31|2.32|2.34|2.34|

---

> ### Author Response · Authors · 2025-11-21
>
> **Weakness1: The technical innovation appears limited, as the method primarily combines previously established techniques of outlier transformation and clustering.**
>
> **A**: While CodeQuant draws inspiration from prior work on outlier smoothing and codebook-based quantization, its contribution goes beyond a simple combination of existing ideas. Each component is specifically designed for MoE architectures and is technically novel in how it adapts or extends earlier concepts. AOS learns rotation matrices by directly optimizing activation quantization error *within* MoE layers, shaping the activation distribution precisely for codebook-based clustering rather than relying on generic rotation schemes. ACCF introduces a tailored output-error minimization objective together with a router-preserving KL term, which is essential for maintaining token–expert consistency during MoE quantization. POG reorganizes weight groups into a more cluster-friendly arrangement while mathematically preserving functional equivalence, enabling lower clustering error that prior quantizers do not address.
>
> Importantly, CodeQuant integrates all of these stages with a LUT-based GEMM kernel that yields practical inference speedups. LUT-based acceleration for low-bit LLMs is itself an emerging area, and has been even less explored in the MoE setting, where routing, expert sparsity, and heterogeneous parameter structures introduce additional complexity. By demonstrating how to adapt LUT-based computation to MoE models, this work makes a meaningful contribution to the efficiency landscape of large sparse architectures.
>
> Finally, the paper strengthens its technical contribution through *real* implementations. It includes an actual CPU LUT-based kernel and a GPU kernel simulation using Accel-Sim, and it further provides a GPU implementation that validates the practicality of the proposed design, as shown by the results in our response to Question 2.

---

> ### Author Response · Authors · 2025-11-21
>
> **Weakness2: The benchmarking and accuracy evaluation are conducted on relatively small models. Since MoE models are typically very large, validating the method on larger-scale models (e.g., Qwen-235B, DeepSeek-R1) would strengthen the claims. Furthermore, evaluation is limited to next-token prediction tasks; testing on more complex reasoning benchmarks (e.g., GSM8K, MATH500) would be more compelling for modern LLM capabilities.**
>
> **A**: We agree that evaluating very large MoE models such as Qwen-235B and DeepSeek-R1, along with more challenging reasoning benchmarks such as GSM8K and MATH500, would further strengthen the work. Consistent with earlier PTQ studies including QuaRot, DuQuant, and SpinQuant, our experiments focus on publicly available models in the tens-of-billions scale, which are more practical to evaluate within the review period. To ensure broad coverage, we selected models with diverse sizes and MoE architectures, namely Phi-mini-MoE-Instruct, Qwen3-30B-A3B, and DeepSeek-V2-Lite. These results collectively demonstrate that CodeQuant is robust across different model families and scales.
>
> At the same time, we are continuing to scale our evaluations and have conducted an A4W4 experiment on Mixtral 8×7B, which is a 47B parameter model. We select Mixtral 8×7B because it shares the same MoE architecture with Phi-mini-MoE while scaling the total parameter count to 47B, which allows us to isolate the effect of model size under an otherwise comparable structure. Under the A4W4 setting, CodeQuant maintains strong performance on Mixtral, achieving perplexity close to the BF16 baseline and delivering competitive accuracy across all QA benchmarks. These results indicate that the benefits of CodeQuant persist at larger scales and that our method remains effective on significantly larger MoE models.
>
> Table1: Mixtral 8x7B A4W4 Embedding-wise CodeQuant Performance
> | Method        | Wiki2      | C4       | A-c       | A-e       | HS        | ML        | PQ        | WG        | Avg       |
> | ------------- | ---------- | -------- | --------- | --------- | --------- | --------- | --------- | --------- | --------- |
> | BF16  | 4.01       | 7.41     | 0.579     | 0.851     | 0.720     | 0.677     | 0.856     | 0.799     | 0.747     |
> | RTN  | 10502.14   | 14045.38 | 0.319     | 0.261     | 0.284     | 0.243     | 0.492     | 0.504     | 0.350     |
> | SqueezeLLM    | 13952.66   | 19725.12 | 0.297     | 0.282     | 0.279     | 0.251     | 0.527     | 0.519     | 0.359     |
> | SmoothQuant   | 77.32      | 96.01    | 0.222     | 0.349     | 0.303     | 0.236     | 0.565     | 0.497     | 0.362     |
> | QuaRot        | 16.79      | 24.29    | 0.348     | 0.570     | 0.512     | 0.286     | 0.708     | 0.560     | 0.497     |
> | **CodeQuant** | **4.65**   | **8.06** | **0.565** | **0.819** | **0.715** | **0.644** | **0.827** | **0.780** | **0.725** |
>
> We further expanded our evaluation to include additional mathematical datasets. Specifically, we ran experiments on GSM8K and MATH500 using DeepSeek-V2-Lite, Qwen3-30B-A3B and Mixtral 8x7B under the A4W4 setting. For GSM8K, we used 8-shot evaluation, and for MATH500, we used 4-shot evaluation, where n-shot indicates that n demonstration examples are provided before the test.
>
> We reproduced the BF16 results to verify the correctness of our evaluation pipeline, and the reproduced numbers are close to those in the DeepSeek-V2 technical report (DeepSeek-AI et al., 2024), with minor differences likely due to prompt formatting. Both GSM8K and MATH500 require multi-step mathematical reasoning and are known to be highly sensitive to quantization. As shown in the table, CodeQuant achieves substantial gains over QuaRot on both benchmarks while staying close to the BF16 upper bound. This demonstrates that CodeQuant preserves reasoning-heavy capabilities even under the A4W4 setting, providing evidence that the method generalizes beyond standard language modeling tasks.
>
> Table2: Math Reasoning Performance on DeepSeek-V2-Lite, Qwen3-30B-A3B and Mixtral 8x7B
> |Model|Method|GSM8K|MATH500|
> |---|---|---|---|
> |DeepSeek-V2-Lite|BF16|0.364|0.121|
> ||QuaRot|0.231|0.093|
> ||**CodeQuant**|**0.330**|**0.108**|
> |Qwen3-30B-A3B|BF16|0.924|0.322|
> ||QuaRot|0.508|0.128|
> ||**CodeQuant**|**0.867**|**0.241**|
> |Mixtral 8x7B|BF16|0.523|0.163|
> ||QuaRot|0.008|0.015|
> ||**CodeQuant**|**0.451**|**0.145**|
>
> **Weakness3: While inference performance is compared, a more detailed breakdown is lacking. It would be beneficial to provide detailed testing parameters and analyze performance separately for the prefill and decoding stages, as they have different computational characteristics (memory bandwidth vs. compute density).**
>
> **A**: We thank the reviewer for the suggestion, since this will strengthen the practical understanding of inference behavior. We provide the requested breakdown, including testing parameters and separate analyses for prefill and decoding, in our detailed response to Question 2 of the rebuttal.

---

> > ### Author Response · Authors · 2025-11-27
> >
> > Dear reviewer yk1W, please let us know if our response has addresses your questions, and feel free to let us know if you have any additional questions. Thank you.

---

### Official Review · Reviewer_JwHU · 2025-11-01

**Soundness:** 2
**Presentation:** 3
**Contribution:** 2
**Rating:** 4
**Confidence:** 3

**Summary:**

The paper proposes CodeQuant, a unified clustering + quantization framework tailored for Mixture-of-Experts (MoE) models. Empirical results on several MoE models show improved perplexity. The paper claims up to 4.15× speedup with negligible accuracy loss.

**Strengths:**

- Addressing outliers in MoE quantization is highly relevant since activation and weight spikes are key bottlenecks in scaling low-bit inference.
- Combining activation smoothing (AOS), column reordering (POG), adaptive clustering (ACCF), and custom LUT kernels forms a complete end-to-end quantization pipeline.

**Weaknesses:**

- The evaluation omits strong baselines such as Duqant and GPTQ.
- Applying learned rotations within MoE layers risks disturbing the routing behavior. Although the paper mentions avoiding direct interference with router inputs, no quantitative analysis or bounds on routing consistency are provided.

**Questions:**

- On which layers or experts does CodeQuant fail? Are there cases where AOS rotation worsens quantization error or diverges?
- How is the trade-off between the number of centroids and quantization precision chosen?
- What is the formal connection between ACCF loss and quantization error？How do you ensure centroid fine-tuning does not simply overfit the calibration set?

---

> ### Author Response · Authors · 2025-11-21
>
> We thank the reviewer for the insightful comments, next we summarize each question and answer them separately:
>
> **Questions1: On which layers or experts does CodeQuant fail? Are there cases where AOS rotation worsens quantization error or diverges?**
>
> **A**: Across all three MoE models, CodeQuant training remained numerically stable and we did not observe divergence or exploding loss. As for numerical illustration, we conduct a new experiment using DeepSeek-V2-Lite under A4W4 setting aiming to illustrate CodeQuant’s impact across layers. Under CodeQuant, ‘A4W4’ corresponds to applying 4-bit linear quantization to activations and clustering weights into $2^4 = 16$ centroids.
>
> We uniformly sample and perform CodeQuant only on a subset of layers and leave the rest of layers quantized using QuaRot, and we discovered a consistent improvement across all our experiments. Meanwhile, it is clear that with more layers quantized using CodeQuant, both the perplexity and accuracy get improved. Specifically, we partition the LLM into three consecutive blocks (layers 1 to 9, 10 to 18, and 19 to 27) and apply the CodeQuant to each block. We then compare the resulting performance with QuaRot under the same bit width. As shown in the following table 1, CodeQuant consistently improves accuracy, demonstrating the robustness of this approach.
>
> Table1:  Layer-wise impact of applying CodeQuant on DeepSeek-V2-Lite (A4W4).
> | Model                                          | Wiki2      | C4    | Accuracy |
> | ---------------------------------------------- | ---------- | ----- | -------- |
> | QuaRot (all layers)                            | 7.75       | 10.75 | 0.641    |
> | CodeQuant on layers 1-9; QuaRot on other       | 7.50       | 10.42 | 0.653    |
> | CodeQuant on layers 10-18; QuaRot on other     | 7.50       | 10.34 | 0.650    |
> | CodeQuant on layers 19-27; QuaRot on other     | 7.46       | 10.35 | 0.650    |
> | CodeQuant on 9 random layers; QuaRot on other  | 7.55       | 10.44 | 0.651    |
> | CodeQuant on 16 random layers; QuaRot on other | 7.40       | 10.24 | 0.658    |
> | CodeQuant (all layers)                         | 7.10       | 9.87  | 0.665    |
>
> In addition, let $Y_{\text{ori}}$ denote the layer outputs produced using FP16 inputs and weights, $Y_{\text{codequant}}$ the layer outputs generated with quantized inputs and clustered weights, and $Y_{\text{quarot}}$ the intermediate results from QuaRot. We observe that $\lVert Y_{\text{ori}} - Y_{\text{codequant}} \rVert^{2}$ is consistently lower than $\lVert Y_{\text{ori}} - Y_{\text{quarot}} \rVert^{2}$, with an average reduction of 0.342 across the calibration set of 512 samples on DeepSeek-V2-Lite.
>
> Moreover, we evaluate the effect of AOS on reducing the quantization error of input activations by comparing $\lVert X_R - Q(X_{R}) \rVert^{2}$ and $\lVert X' - Q(X') \rVert^{2}$, where $X_{R} = X R$ with (R) being the learned rotation matrix, and $X' = X V$ with $V$ being a randomly selected (non-learned) rotation matrix. The table below shows the quantization error statistics across all the blocks within DeepSeek-V2-Lite. As shown in the table below, AOS consistently reduces quantization error across all decoder blocks compared to random rotation. These results demonstrate that the improvement is systematic rather than block-specific.
>
> Table2: Quantizaiton Error Comparison (Block)
> | Method                                          | Mean                                     | Variance | 25 Percentile | Median | 75 Percentile | Max   |
> | ----------------------------------------------- | ---------------------------------------- | -------- | ------------- | ------ | ------------- | ----- |
> | Random Rotation ($\lVert X' - Q(X') \rVert^{2}$) | 1.081                                    | 0.374    | 1.043         | 1.051  | 1.131         | 3.544 |
> | AOS ($\lVert X_R - Q(X_{R}) \rVert^{2}$)          | 0.693                                    | 0.126    | 0.629         | 0.659  | 0.754         | 1.961 |
>
> In addition, we also compare linear layer-wise (projection-wise) quantization errors on DeepSeek-V2-Lite. The logic is similar to the previous table. Our experiment's results show the same pattern as the block-wise result, that AOS will reduce quantization error compared to random rotation.
>
> Table3: Quantizaiton Error Comparison (Porjection Layer)
> | Method                                          | Mean                                          | Variance | 25 Percentile | Median | 75 Percentile | Max   |
> | ----------------------------------------------- | --------------------------------------------- | -------- | ------------- | ------ | ------------- | ----- |
> | Random Rotation ($\lVert X' - Q(X') \rVert^{2}$) | 0.064                                         | 0.004    | 0.025         | 0.063  | 0.078         | 0.436 |
> | AOS ($\lVert X_{R} - Q(X_{R}) \rVert^{2}$)          | 0.055                                         | 0.003    | 0.022         | 0.057  | 0.065         | 0.387 |

---

> ### Author Response · Authors · 2025-11-21
>
> **Question2. How is the trade-off between the number of centroids and quantization precision chosen?**
>
> **A**: There is no trade-off between the number of centroids and the quantization precision because these two quantities cannot be varied independently. If we have an N-bit codebook then we have exactly $2^N$ centroids. In our implementation, we choose 16 centroids following the standard 4-bit PTQ setting, where a 4-bit codebook corresponds to $2^4 = 16$ entries. Importantly, our method is not brittle to this choice: even when reducing the number of centroids, CodeQuant remains stronger than prior clustering-based approaches. As shown in our ablations (Table 5 in the revision of paper), decreasing the codebook size (e.g., 8 or 4 centroids) still yields higher accuracy than SqueezeLLM under the same codebook budget, demonstrating robustness of CodeQuant.
>
> Additionally, to study the tradeoff between the number of weight centroids and the activation quantization precision, we evaluate performance under different combinations of activation bit width and centroid count. Specifically, we test three configurations on DeepSeek-V2-Lite:
>
> - A10W2, where activations use 10-bit quantization and weights use 4 centroids;
> - A8W4, where activations use 8-bit quantization and weights use 16 centroids;
> - A6W6, where activations use 6-bit quantization and weights use 64 centroids;
>
> This comparison allows us to analyze the balance between activation precision and weight expressiveness under varying quantization regimes. We observe that A6W6 provides slightly better accuracy than both A8W4 and A10W2, and A10W2 achieves the lowest accuracy due to the extremely limited number of centroids ($2^2 = 4$).
>
> Table1: CodeQuant Comparison with Different Budget
> | Model            | Bits  | Wiki2      | C4    | Accuracy |
> | ---------------- | ----- | ---------- | ----- | -------- |
> | DeepSeek-V2-Lite | A10W2 | 9.67       | 13.45 | 0.580    |
> |                  | A8W4  | 6.84       | 9.51  | 0.679    |
> |                  | A6W6  | 6.72       | 9.35  | 0.681    |
>
> **Question3. What is the formal connection between ACCF loss and quantization error？How do you ensure centroid fine-tuning does not simply overfit the calibration set?**
>
> **A**: According to Equation 4, ACCF minimizes the L2 difference between each layer’s output under full-precision inputs and weights and the output produced using quantized activations and clustered weights. In the ideal case, a zero loss would guarantee that CodeQuant’s output matches the original full-precision output, resulting in no accuracy drop. Additionally, we introduce a term that minimizes the discrepancy in routing decisions before and after applying CodeQuant, which has been shown to improve the final accuracy based on the results of Table 3 (Table 5 in the revision version).
>
> To verify that ACCF does not overfit to the calibration set, we examined cases where overfitting is most likely to occur. In all standard configurations used in the paper, increasing the number of calibration epochs does not lead to degradation, and the downstream accuracy remains stable. Overfitting emerges only under deliberately extreme conditions. For example, when the calibration set is reduced to just 16 WikiText-2 samples and trained extensively, the model clearly overfits: perplexity rises to 282.80 on WikiText-2 and 2825.76 on C4, and accuracy drops to 0.364. The disproportionately large degradation on C4 is expected, as the model becomes over-specialized to the tiny in-domain WikiText-2 subset and fails to generalize out of distribution—indicating classic overfitting behavior. Importantly, this effect disappears once the calibration set is even moderately increased. Using 64 samples or more is already sufficient to avoid overfitting (which can improve the perplexity of 7.10 on WikiText-2 and 9.87 on C4). These results indicate that ACCF is stable in practical settings and only overfits when the calibration set is unrealistically small.

---

> ### Author Response · Authors · 2025-11-21
>
> **Weakness1: The evaluation omits strong baselines such as Duqant and GPTQ.**
>
> **A**:We compare CodeQuant with GPTQ in the same setting. Since GPTQ does not apply any rotational transformation, it incurs a large input-activation quantization error when using 4-bit precision. Our CodeQuant (in Table 1 of the paper) outperforms GPTQ in accuracy by 12%–13% for both DeepSeek-V2-Lite and Qwen3-30B-A3B, and also lowers perplexity on both WikiText-2 and C4.
>
> Meanwhile, in the revison, we include DuQuant as a rotation-based PTQ baseline under the A4W4 configuration (Table 2). As shown in the table, our Hadamard-enabled variant, CodeQuant, denoted as $\text{CodeQuant}_{had}$, consistently outperforms DuQuant across all QA tasks on all evaluated models, achieving a higher average accuracy and lower perplexity on both WikiText2 and C4. These results demonstrate that our method remains highly competitive even when compared with strong rotation-based approaches.
>
> Table1: Evaluation of PTQ Quantization Methods on DeepSeek-V2-Lite, Qwen3-30B-A3B and Mixtral 8×7B
> | Model   | Method        | Wiki2      | C4    | A-c   | A-e   | HS    | ML    | PQ    | WG    | Avg   |
> |----------|----------------|-----------|--------|-------|-------|-------|-------|-------|------|--------|
> | DeepSeek-V2-Lite | GPTQ           | 14.10     | 16.62  | 0.383 | 0.578 | 0.612 | 0.347 | 0.707 | 0.615 | 0.541 |
> |          | DuQuant        | 8.43      | 11.94  | 0.455 | 0.708 | 0.715 | 0.400 | 0.775 | 0.693 | 0.658 |
> |          | CodeQuant_had  | 8.16      | 11.38  | 0.445 | 0.723 | 0.727 | 0.454 | 0.782 | 0.644 | 0.666 |
> | Qwen3-30B-A3B    | GPTQ           | 17.79     | 25.56  | 0.345 | 0.641 | 0.628 | 0.579 | 0.707 | 0.510 | 0.565 |
> |          | DuQuant        | 13.52     | 20.10  | 0.472 | 0.662 | 0.687 | 0.654 | 0.739 | 0.606 | 0.637 |
> |          | CodeQuant_had  | 12.69     | 19.89  | 0.477 | 0.697 | 0.691 | 0.679 | 0.739 | 0.635 | 0.653 |
> | Mixtral 8×7B  | DuQuant        | 6.13      | 9.94   | 0.519 | 0.775 | 0.676 | 0.583 | 0.832 | 0.764 | 0.691 |
> |          | CodeQuant_had  | 5.95      | 9.86   | 0.536 | 0.808 | 0.693 | 0.670 | 0.848 | 0.728 | 0.714 |
>
>
> **Weakness2: Applying learned rotations within MoE layers risks disturbing the routing behavior. Although the paper mentions avoiding direct interference with router inputs, no quantitative analysis or bounds on routing consistency are provided.**
>
> **A**: The paper shows that the rotation matrix (R) is applied jointly to activations and weights:
>
> 1. Activations: ($X \rightarrow XR$)
> 2. Weights:  ($W \rightarrow R^{\top} W$)
>
> Because $RR^\top = I$, we have $\phi(X R R^\top W_{\text{gate}}) \odot (XR R^\top W_{\text{up}}) W_{\text{down}}= (\phi(X W_{\text{gate}}) \odot X W_{\text{up}}) W_{\text{down}}$. This shows that the expert outputs, including the router’s linear transformation, remain identical to the output before, meaning the rotation is output-invariant, as shown in Section 3.1. Thus, at the mathematical level, the router logits and decisions are unaffected by the rotation itself.
>
> Quantizing the inputs and clustering the weights can shift the router logits and change the expert assignments. To reduce this effect, the KL term serves as a routing consistency regularizer that encourages the quantized router to remain close to the original token to expert distribution, as described in Equation 4 of the paper. For quantitative evidence, we report the router change rate, which is the percentage of tokens whose top one expert differs from the original, on Qwen3-30B-A3B and DeepSeek-V2-Lite. The KL regularization clearly reduces these routing changes. As shown in the ablation studies in Table 3 (Table 5 in the revised version), including this additional term also leads to improved accuracy.
>
> Table2: Router Change Rate Comparison
> | Model                      | Method | Change Rate (%) |
> | -------------------------- | ------ | --------------- |
> | DeepSeek-V2-Lite           | QuaRot | 41.47           |
> |    |  CodeQuant w/ KL divergence |  22.82           |
> | Qwen3-30B-A3B              | QuaRot | 72.15           |
> |   |   CodeQuant w/ KL divergence |  59.58            |

---

> > ### Author Response · Authors · 2025-11-27
> >
> > Dear reviewer JwHU, please let us know if our response has addressed your questions, and please let us know if there are more questions, thank you.

---

### Comment · Area_Chair_URuQ · 2025-11-24

Dear Reviewers,

**We kindly encourage you to review and respond to the authors’ rebuttals**. Your timely feedback is important for ensuring a fair and thorough review process. Thank you for your contributions to ICLR 2026.

AC

---

### Author Response · Authors · 2025-12-02
**General Response**

We sincerely thank all reviewers for their insightful and constructive feedback. We are encouraged that reviewers recognized CodeQuant's contributions to efficient MoE quantization, particularly our novel co-design approach and practical system implementation. In response to the valuable suggestions, we have substantially strengthened our evaluation and clarified technical details. **All core findings remain valid and are further reinforced by the new experiments.**

**Major Revisions Addressing Reviewer Concerns:**

**1. Comprehensive Baseline Comparisons and Expanded Evaluation (Reviewers JwHU, yk1W, BbsM, 8Fns)**

Following multiple requests for stronger baselines and more challenging benchmarks, we have:

- **Added state-of-the-art baselines**: Experimented with GPTQ[1] and found that CodeQuant consistently outperforms it on both DeepSeek-V2-Lite and Qwen3-30B-A3B. We further included DuQuant[2] and SpinQuant[3] in the comparison (revised Table 2): $\text{CodeQuant}_{had}$ surpasses DuQuant (0.653 vs 0.637 on Qwen3, 0.666 vs 0.658 on DeepSeek-V2-Lite) and outperforms SpinQuant (0.653 vs 0.590 on Qwen3, 0.666 vs 0.647 on DeepSeek-V2-Lite).
- **Evaluated on reasoning benchmarks**: Added GSM8K[4] (8-shot) and MATH500[5] (4-shot) as requested. CodeQuant preserves reasoning capability (0.867 vs 0.924 on GSM8K for Qwen3) while QuaRot degrades severely (0.508). The same pattern is also observed for MATH500.
- **Scaled to larger models**: Validated on Mixtral 8×7B (47B parameters), achieving 0.725 average accuracy vs QuaRot's 0.497, confirming scalability.

**2. Real GPU Implementation Replacing Simulation (Reviewers yk1W, 8Fns, BbsM)**

We now provide comprehensive real hardware validation:

- **NVIDIA A100 measurements**: CodeQuant achieves 2.48× speedup over FP16 baseline.
- **Detailed performance breakdown**: Prefill/decode analysis shows speedup increases with sequence length (2.23× → 2.35× from 128 to 4096 tokens)
- **Memory efficiency**: 3.42× reduction with confirmed zero runtime overhead
- **Preprocessing costs**: AOS takes 15-30 minutes, ACCF takes 30-110 minutes on H100 (one-time offline cost)

**3. Quantitative Analysis of MoE-Specific Concerns (Reviewers JwHU, yk1W)**

Addressing concerns about router disruption and quantization impact:

- **Router consistency metrics**: KL regularization reduces routing change rate from 41.47% to 22.82% (DeepSeek-V2-Lite) and 72.15% to 59.58% (Qwen3)
- **KL Divergence impact**: We add more experiment results on different models, and the detailed ablation confirmed the effectiveness of KL divergence. (Table 5, revised)

**4. Clarified Technical Novelty Beyond Existing Methods (Reviewer BbsM)**

We better articulated CodeQuant's key differences from existing rotation-based PTQ methods (SpinQuant and OSTQuant):

- **Core distinctions**: Unlike SpinQuant/OSTQuant which use end-to-end fine-tuning with global KL objectives, CodeQuant employs (1) per-layer MSE losses for better layer-wise reconstruction, and (2) MoE-specific router consistency preservation through KL regularization - critical for maintaining token-expert assignments.
- **Empirical validation**: CodeQuant outperforms SpinQuant (0.653 vs 0.590 on Qwen3-30B-A3B) and OSTQuant (0.712 vs 0.661 even when fine-tuning only 3 layers given the resource constraint), demonstrating superior performance with lower computational cost.

**5. Corrected Presentation and Added Missing Details (Reviewer 8Fns)**

- **Table correction:** Fixed all table formatting errors while preserving the correct numerical values.
- **Clear Overhead Explanation:** Mathematically and explicitly explain that our method (AOS and POG) will not cause additional computational overhead. During inference, the LUT overhead is further analyzed in Comment 2.
- **More Experiment Details:** We provide the detailed preprocessing times requested by the reviewer in our revised paper.
---
[1] Frantar, Elias, et al. "Gptq: Accurate post-training quantization for generative pre-trained transformers." arXiv preprint arXiv:2210.17323 (2022).
[2] Lin, Haokun, et al. "Duquant: Distributing outliers via dual transformation makes stronger quantized llms." Advances in Neural Information Processing Systems 37 (2024): 87766-87800.
[3] Liu, Zechun, et al. "Spinquant: Llm quantization with learned rotations." arXiv preprint arXiv:2405.16406 (2024).
[4] Cobbe, Karl, et al. "Training verifiers to solve math word problems." arXiv preprint arXiv:2110.14168 (2021).
[5] Hendrycks, Dan, et al. "Measuring massive multitask language understanding." arXiv preprint arXiv:2009.03300 (2020).

---

### Meta-Review · Area_Chair_n3A4 · 2026-01-08

**Summary:**

This paper introduces CodeQuant, a unified quantization-and-clustering framework for efficient deployment of Mixture-of-Experts (MoE) large language models under low-precision constraints (W4A4). The work combines three main technical contributions: (1) Activation-oriented Outlier Smoothing (AOS) using learnable rotation matrices, (2) Adaptive Weight Clustering with Centroid Finetuning (ACCF) with MoE-specific KL regularization, and (3) Permutation-Invariant Outlier Grouping (POG) for improved clustering. The authors demonstrate improved accuracy over existing quantization methods (QuaRot, SmoothQuant, SqueezeLLM) on MoE models ranging from 3B to 47B parameters, with inference speedups of 2.48× on real A100 GPUs and up to 4.15× on CPUs.

Initial reviewer scores ranged from 4/10 to 6/10, with concerns about limited baselines, simulation-only GPU evaluation, unclear technical novelty, and missing analysis on reasoning tasks and larger models. The authors provided a comprehensive rebuttal that substantially strengthened the paper through:

1. Addition of strong baselines (GPTQ, DuQuant, SpinQuant)

2. Real GPU implementation on A100 replacing simulation results

3. Evaluation on larger models (Mixtral 8×7B, 47B parameters)

4. Mathematical reasoning benchmarks (GSM8K, MATH500)

5. Quantitative analysis of router consistency and MoE-specific impacts

Based on the discussion, the strengths and weaknesses of this paper are:

Strengths:

1.	Comprehensive evaluation: Three MoE models (3B-47B parameters), multiple quantization settings, both standard and reasoning benchmarks.

2.	Practical impact: Real hardware validation showing 2.48× GPU speedup and 4.15× CPU speedup with maintained accuracy.

3.	MoE-specific contributions: Router consistency preservation through KL regularization, demonstrated to reduce routing changes by ~40-50%.

4.	Thorough rebuttal: Authors addressed all major concerns with substantial new experiments and analysis.

5.	Reproducibility: Detailed methodology, public datasets, fixed seeds; code release promised upon acceptance.

Weaknesses:

1.	Limited novelty: Core techniques (rotation, clustering, permutation) are adaptations of existing methods

2.	Hardware gap: Simulated results assume architectural changes not present in current GPUs.

3.	Incremental improvements: While consistent, accuracy gains over strong baselines (DuQuant, SpinQuant) are often modest (1-2%).

Why Accept: Despite incremental novelty, the paper makes solid contributions to an important problem (efficient MoE deployment). The comprehensive evaluation, real hardware implementation, and MoE-specific adaptations demonstrate practical value. The work represents competent engineering and empirical validation that advances the state of practice, even if not providing fundamental algorithmic breakthroughs.

**Reviewer Concerns:**

Addressed by Rebuttal:

1. Missing Strong Baselines (Reviewers JwHU, BbsM):

•	Authors added comparisons with GPTQ, DuQuant, and SpinQuant.

•	CodeQuant consistently outperforms: 0.653 vs 0.637 (DuQuant) and 0.653 vs 0.590 (SpinQuant) on Qwen3-30B-A3B.

•	Results demonstrate competitive performance against state-of-the-art rotation-based PTQ methods.

2. GPU Evaluation Only via Simulation (Reviewers yk1W, 8Fns, BbsM):

•	Authors provided real A100 GPU measurements showing 2.48× speedup over FP16.

•	Detailed prefill/decode analysis: speedup increases with sequence length (2.23× → 2.35×).

•	Memory efficiency: 3.42× reduction with zero runtime overhead.

•	Preprocessing costs reported: AOS 15-30 min, ACCF 30-110 min on H100 (one-time offline cost).

3. Limited Model Scale and Task Coverage (Reviewers yk1W, 8Fns):

•	Scaled evaluation to Mixtral 8×7B (47B parameters): 0.725 vs QuaRot's 0.497 average accuracy.

•	Added GSM8K (8-shot) and MATH500 (4-shot) reasoning benchmarks.

•	CodeQuant preserves reasoning: 0.867 vs 0.924 BF16 on GSM8K for Qwen3; QuaRot severely degrades to 0.508.

4. MoE-Specific Quantitative Analysis (Reviewers JwHU, yk1W):

•	Router consistency metrics: KL regularization reduces routing change rate from 41.47% to 22.82% (DeepSeek-V2-Lite).

•	Layer-wise impact analysis shows consistent improvement across all layers.

•	Quantization error analysis: AOS reduces error by mean 0.342 compared to random rotation.

5. Technical Novelty Clarification (Reviewer BbsM):

•	Distinguished from SpinQuant/OSTQuant: per-layer MSE vs global KL objectives.

•	MoE-specific router consistency preservation through KL regularization.

•	Empirical validation: outperforms SpinQuant (0.653 vs 0.590) and OSTQuant (0.712 vs 0.661).

Remaining Concerns:

1. Incremental Nature of Contributions: While the authors clarified technical differences from prior work, the core components (learnable rotations, clustering, permutation) build heavily on established techniques. The main novelty lies in:

•	MoE-specific adaptations (router KL regularization).

•	Integration into a unified framework with hardware co-design.

•	Practical system implementation.

The contribution is solid but somewhat incremental rather than fundamentally novel.

2. Hardware Assumptions: The simulated GPU results (2.63× speedup) assume architectural modifications (optimized LUT operations, increased shared memory banks) not present in current hardware. Real GPU speedups (2.48×) are more modest. While the authors acknowledge this, the gap raises questions about practical near-term deployment.

3. Limited Analysis on When POG Helps: POG provides clear benefits under extreme compression (A4W4 Block-wise) but minimal or negative impact at A8W4. The authors note this occurs when models are already close to BF16 baseline, but deeper analysis of when permutation is beneficial would strengthen the work.

4. Preprocessing Costs: While reported (15-110 minutes on H100), the paper could better contextualize these one-time costs relative to deployment scenarios and compare against baseline methods' calibration times.

**Reviewer Scores:**

Based on the comprehensive rebuttal and substantial improvements:

•	Reviewer JwHU (initially 4): Likely would increase to 6 given strong responses on baselines, layer-wise analysis, and MoE-specific quantitative metrics.

•	Reviewer yk1W (initially 4): Likely would increase to 6 given real hardware validation, larger model evaluation, and reasoning benchmarks.

•	Reviewer 8Fns (initially 6): Would likely maintain or increase to 6-7 given comprehensive responses on all weaknesses.

•	Reviewer BbsM (initially 6): Likely would maintain 6 or increase to 7 given clarification of novelty and real hardware results.

---

### Decision · Program_Chairs · 2026-01-26

Accept (Poster)